



# Glacial-interglacial sea water isotope change near the Chilean Margin as reflected by δ²H of C37 alkenones.

Katrin Hättig[1], Devika Varma[1], Stefan Schouten[1,2], Marcel T. J. van der Meer[1]

[1] Department of Marine Microbiology and Biogeochemistry, NIOZ Royal Netherlands Institute for Sea Research, Texel, the Netherlands

[2] Department of Earth Sciences, Faculty of Geosciences, Utrecht University, Utrecht, the Netherlands

*Correspondence to*: Katrin Hättig (katrin.haettig@nioz.nl)

**Abstract.** Hydrogen isotopic compositions of C37 long chain alkenones ($\delta^2H_{C37}$) have been proposed as a proxy for past seawater salinity and applied in several regions. However, previous studies were based solely on a single core and often suggested unlikely large changes in salinity. Here we present a new $\delta^2H_{C37}$ record, in combination with oxygen isotopes of benthic foraminifera from the same samples, from a sediment core from the Chilean Margin (ODP site 1235). The observed negative shift in $\delta^2H_{C37}$ of 20‰ during deglaciation was identical to that of the nearby located, but deeper, ODP core 1234, suggesting a regionally consistent shift in $\delta^2H_{C37}$. This shift translates into a hydrogen isotope shift of the surface seawater ($\Delta\delta^2H_{SW}$) of ca. 14‰, similar to glacial-interglacial reconstructions based on other $\delta^2H_{C37}$ records. The reconstructed bottom seawater oxygen isotope change based on benthic foraminifera during the last deglaciation is approximately 0.8‰, in line with previous studies. When translated into $\Delta\delta^2H_{SW}$ using the modern open-ocean waterline this would suggest a shift of ca. 5‰, smaller than the reconstructed surface seawater shift based on alkenones. The higher change in surface $\delta^2H_{SW}$ suggests that the surface water experienced more freshening during the Holocene than bottom waters either due to increased freshwater input or reduced evaporation, or a combination of the two.

## 1 Introduction

The isotopic composition of seawater ($\delta^{18}O_{sw}$/ $\delta^2H_{sw}$) reveals crucial information about (sea)water physicochemical properties which could improve our understanding of past global ocean current dynamics and changes therein. Several oceanographic factors can cause shifts in the isotopic composition of seawater: for example, the bottom sea water isotopic composition depends on the source area of the water and reflects the history of advection associated transport, mixing, upwelling and downwelling of ocean currents (Rohling and Cooke, 1999). Regional factors affecting the sea surface water isotopic composition are the evaporation-precipitation balance, river water input, iceberg melting, and the local climate regime (Rohling and Cooke, 1999). On a more global level, land ice formation is an important process which affects oxygen and hydrogen isotopes of sea water: during glacial periods large amounts of water with relatively low oxygen and hydrogen isotope values are stored as ice on land, leaving the sea water more enriched in the heavier isotopes (e.g. Schrag et al., 2002). During warmer interglacials, the decrease in global ice volume leads to the formation of low density fresh water with relatively low $\delta^{18}O$ and



$\delta^2$H values which enters the surface ocean (Rohling and Bigg, 1998). Since evaporation and precipitation affect sea water isotopes and salinity in a similar way, seawater isotopes have a close relationship with the salinity of the ocean (Rohling 2007; Craig 1961; Craig and Gordon 1965). Therefore, sea water isotopes are often reconstructed and used to infer past salinity changes (e.g. Lamy et al., 2002; Schrag et al., 2002; Rousselle et al., 2013).

There are several methods available to reconstruct the isotopic composition of sea water. The stable oxygen isotopic composition of foraminifera ($\delta^{18}O_{Foram}$) is a function of the oxygen isotope composition of ambient sea water ($\delta^{18}O_{sw}$) and calcification temperature (Bemis et al., 1998; Epstein et al., 1953; Shackleton, 1974) with a minor effect of carbonate ion concentration (Spero et al., 1997). Temperature-corrected $\delta^{18}O_{foram}$ is thus assumed to reflect $\delta^{18}O_{sw}$ though with some uncertainty (Duplessy et al., 1991; Rostek et al., 1993). Generally, temperature corrections are done based on temperature

proxies derived from the same foraminifera, such as Mg/Ca (e.g. Billups et al., 2002, 2003; Lear et al., 2000, 2004, 2015; Martin et al., 2002) or clumped isotopes ($\Delta_{47}$) (e.g. Petersen et al., 2015). In some cases, temperature proxies based on organic geochemical indices, such as $U^{K'}_{37}$ and $TEX_{86}$ (Brassell et al., 1986; Prahl et al., 1987; Schouten et al., 2003), are used to correct for the temperature (Rostek et al., 1993, 1997). Several studies based on $\delta^{18}O_{Foram}$–records and modelling estimate a global average change of 0.8-1.1‰ in the oxygen isotope ratio of the global sea water during the last deglaciation (e.g. Adkins

et al., 2002; Duplessy et al., 2002; Oba et al., 2004; Waelbroeck et al., 2002). To derive salinity from local $\delta^{18}O_{sw}$ estimates, i.e. those corrected for global $\delta^{18}O_{sw}$ changes, requires knowledge on the regional relationship between $\delta^{18}O_{sw}$ and the salt content (LeGrande and Schmidt, 2006; Rohling, 2007). However, it is uncertain whether this relationship was similar in the past as it is today.

The hydrogen isotopic composition of sea water is more difficult to reconstruct. One way is to measure the hydrogen isotopic

composition of pore fluids in deep sea sediments. Those indicate a change of 6-9‰ in the $\delta^2H_{sw}$ composition of the bottom water during the last glacial–interglacial change (Schrag et al., 2002). Another potential proxy is based on the combined $C_{37:2}$ and $C_{37:3}$ alkenones ($\delta^2H_{C37}$), produced by haptophyte algae. Culture studies show that the hydrogen isotopic fractionation of phototrophic organisms depends not only on the hydrogen isotopic composition of sea water but also on salinity (M'Boule et al., 2014; Sachs et al., 2016; Schouten et al., 2006; Weiss et al., 2017a; Zhang et al., 2009; Zhang et al., 2007). However, the

$\delta^2H_{C37}$–salinity response varies significantly between haptophyte species (M'Boule et a., 2014; Sachs et al., 2016; Schouten et al., 2006), as well as in the environment (Gould et al., 2019; Häggi et al., 2015; Mitsunaga et al., 2022; Weiss et al., 2019b). These differences in response in $\delta^2$H to salinity makes quantitatively constraining past salinity changes difficult. The application of $\delta^2H_{C37}$ to sediment cores show a shift of 16-25‰ (when corrected for ice volume 9-18‰) for the last deglaciation depending on the site (Kasper et al., 2014; Pahnke et al., 2007; Petrick et al., 2015; Simon et al., 2015; Weiss et al., 2019a).

This large glacial interglacial shift in $\delta^2H_{C37}$ suggests large salinity changes (5 to 19 psu) when using culture and especially core top calibrations, much larger than expected. Therefore, Weiss et al. (2019b) concluded a higher paleo sensitivity of $\delta^2H_{C37}$ to salinity changes than observed in modern day environments and cultures.

A different approach would be to directly correlate $\delta^2H_{C37}$ with $\delta^2H_{SW}$. Gould et al. (2019) observed a statistically significant $\delta^2H_{C37} – \delta^2H_{SW}$ relationship with open-ocean suspended particulate organic matter (SPOM) and Mitsunaga et al. (2022) showed



a statistically identical relationship based on core top sediments. This suggests that in the natural environment, the effects of factors like salinity, species composition (e.g. Chivall et al., 2014; M'Boule et al., 2014) and light and nutrients (Sachs et al., 2015; van der Meer et al., 2015) on stable hydrogen isotope fractionation during biosynthesis may be less important than the isotopic composition of sea water. This observation allows direct inference of $\delta^2H_{SW}$ from $\delta^2H_{C37}$ without any additional correction, though this approach has not been applied yet to reconstruct past $\delta^2H_{SW}$. Mitsunaga et al. (2022) did predict a

"global" LGM-Modern change of 12.7‰ in the hydrogen isotopic composition of alkenone based on several assumptions. Similar to $\delta^{18}O_{SW}$, reconstructed $\delta^2H_{SW}$ can then be converted into salinity using modern day correlations between surface $\delta^2H_{SW}$ and salinity  (LeGrande and Schmidt, 2006; Rohling, 2007).

To address the issue of local versus regional variability of $\delta^2H_{C37}$ we generated a $\delta^2H_{C37}$ and benthic $\delta^{18}O_{Foram}$ record of ODP core 1235 from the Chilean Margin (Fig. 1) and compared this with the previously published hydrogen isotope (Weiss et al.,

2019b) and temperature records (de Bar et al., 2018) from nearby ODP core 1234 which is only ~12 km away though at a much greater water depth (489 m versus 1015 m). Furthermore, in contrast to Weiss et al. (2019b) we reconstructed $\delta^2H$ of surface seawater and compared this to bottom seawater oxygen isotopes inferred from benthic foraminifera and used both to constrain relative salinity change during the deglaciation.

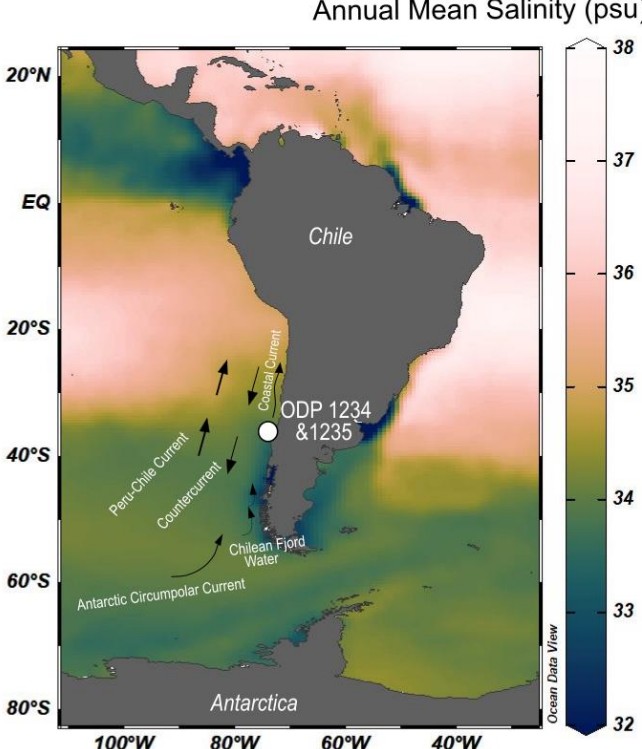

**Figure 1:** Map of sea surface salinity (from Zweng et al., 2018, scientific colour map from Crameri et al., 2022) showing the sediment core

locations of the neighbouring cores ODP 1234 and ODP 1235. The core location is influenced by two primary ocean currents: the Northward Peru-Chile Current and the Chilean Coastal Current, as well as the southward Peru-Chile Countercurrent, Those are fed by the Antarctic Circumpolar waters and Chilean Fjord waters can also enter from the southeast the system.



## 2 Material and Methods

### 2.1 Geographic Setting

The ODP core 1235 was retrieved at 36.16°S, 73.57°W at a shallow water depth of 489 m (Fig. 1). The core is about 12 km away from the deeper located neighboring ODP 1234 core (36.22°S, 73.68°W, water depth 1015 m) in the southeast Pacific Ocean (Mix et al., 2003) (Fig. 1). These cores are located on the Chilean Margin which is influenced by surface and deep-water currents which cause upwelling stimulating local productivity (Bakun, 1989). The setting is dynamic with south westerly winds and freshwater runoff from the continent. The Antarctic Circumpolar Current (ACC) is feeding into the Peru-Chile

Countercurrent and transports cold saline waters. Isotopically depleted water can potentially be added by the Chilean Fjord water which feeds the Chile Coastal Current (Lamy et al., 2002; Strub et al., 1998; De Bar et al., 2018). The present-day annual mean sea surface salinity in the study area is 33.8±0.2 psu (Zweng et al., 2018).

### 2.2 Sampling and age model

Sediment samples of core ODP 1235 were split into two subsamples: Approximately 6 g of wet sediment was used for

foraminiferal oxygen isotope analysis ($\delta^{18}O_{Foram}$) and approximately 10 g of wet sediment was used for organic proxy analysis i.e., $U^{K'}_{37}$, $TEX_{86}$ (see Varma et al., *under revision*) and $\delta^2H_{C37}$ (this study). Samples were taken every 5-10 cm to cover a resolution of <2 ka a total of 27 samples were analysed for hydrogen isotopes.

The age-depth models for ODP 1234 and ODP 1235 were constructed based on the radiocarbon ages of mixed benthic foraminifera (Muratli et al., 2010). Additionally, three radiocarbon measurements from benthic foraminifera species *Uvigerina*

from ODP 1235 were performed using the Mini Carbon Dating System (MICADAS) at the Alfred Wegener Institute (AWI; Table S1). The oldest tie-point for both cores is the Laschamp event at 41.000 years, based on magnetic susceptibility correlation between the cores ODP 1233, ODP 1234 and ODP 1235 (Mix et al., 2003; Table S1). Since the latter tie point of 41 ka, with 51 meters and 26 meters composite depth for ODP 1234 and 1235, respectively, is a rough estimation (Mix et al., 2003), the data points in the records between 30 – 40 ka likely have an uncertainty of around ±1 ka assuming a constant

sedimentation rate. Radiocarbon ages were converted into calendar years using Marine09 data set (Reimer et al., 2009) with the reservoir correction given by (Muratli et al., 2010). The core chronology was constructed with the statistical package Bacon (Blaauw and Christeny, 2011) using a Bayesian approach. ODP 1234 has a mean sedimentation rate of 20 cm.ka$^{-1}$ and ODP 1235 a rate of 5 cm.ka$^{-1}$.

### 2.3 Analysis of benthic foraminifera and long-chain alkenones

Stable oxygen isotopes were measured on the benthic foraminifera species *Uvigerina* with an automated carbonate device (Kiel IV, Thermo Scientific) connected to a Thermo Finnigan MAT 253 dual-inlet isotope-ratio mass spectrometer. The sample sizes were ca. 20 µg and contained 2-5 foraminifera shells (>150 µm). Replicate analyses of $\delta^{18}O$ of the same sample were





averaged with average differences being 0.016-0.2‰. NBS 19 limestone was used as isotope calibration standard and for drift detection and correction a second standard, NFHS-1, was used. The standard deviation is within 0.1‰ for $\delta^{18}O$. All carbonate isotope data are given in per mill (‰) relative to V-PDB (Vienna – Pee Dee Belemnite). Calculated $\delta^{18}O$ seawater values are given in per mill (‰) relative to V-SMOW (Vienna – Standard Mean Ocean Water) with the accepted conversion value of 0.27‰ (Hut, 1987; Pearson, 2012).

The sediment split for organic analysis was extracted as described by Varma et al. (*under revision*). Briefly, sediments were first freeze dried and then extracted using an Accelerated Solvent Extractor. Solvent from the total extract was first removed using a Turbovap and the last remaining water, if present, was removed using a small sodium sulfate column to obtain the total lipid extract (TLEs). The TLEs were separated into three fractions over a small aluminum oxide ($Al_2O_3$) column using hexane/ dichloromethane 9:1 (v:v) to elute first the apolar fraction, hexane/ DCM 1:1 (v:v) to elute the ketone (alkenone) fraction, and dichloromethane/ methanol 1:1 (v:v) to elute the polar fraction. The ketone fractions were used for compound-specific hydrogen isotope ratio measurements of alkenones, after being measured using a gas chromatograph coupled to a flame ionization detector (GC-FID) to determine the correct concentration and complexity of the fraction. Hydrogen isotope ratios were measured in duplicate using a gas chromatograph coupled to a Thermo Delta V isotope ratio mass spectrometer via high-temperature conversion reactor (Isolink I) and Conflo IV. The GC was equipped with an RTX-200 60 m column following Weiss et al. (2019). We report the hydrogen isotope ratio, relative to V-SMOW, of the individual alkenones $C_{37:3}$ and $C_{37:2}$ and the integrated $C_{37}$ $\delta^2H$ ratios determined by manual peak integration of the combined $C_{37:3}$ and $C_{37:2}$ peaks (van der Meer et al., 2013). The $H_3+$ factor was measured daily prior to running samples and shifted in time never more than 0.5 ppm/nA per day. The performance and stability of the instrument was monitored by measuring an n-alkane mixture standard, Mix B (A. Schimmelmann, Indiana University) at the start of each day. Samples were only run when the average difference and standard deviation between online and offline values was less than 5‰. With each sample squalene and a C30 n-alkane were coinjected to monitor the system performance with measured values ranging from -173 ±5‰ and -78 ±4‰. The offline determined values are −170 ±4‰ for Squalene and −79 ±5‰ for the C30 n-alkane.

**2.4 Modern open-ocean isotopic composition of sea water**

The relationship between $\delta^{18}O$-$\delta^2H$ in the water-atmosphere environment can be described with the meteoric water line (MWL) and is close to $\delta^2H = 8 \times \delta^{18}O + 10$ according to Craig (1961) and Craig et al. (1965). To describe the relationship in surface ocean waters and investigate the potential for palaeoceanographic applications Rohling (2007) correlated measured $\delta^{18}O$ and $\delta^2H$ of sampled surface ocean waters. We updated the dataset of Rohling (2007) with data from the Waterisotope Database (2022) managed by Dr. G. Bowen (University of Utah) and published data from Gould et al. (2019), Weiss et al. (2019a) and Srivastava et al. (2010) (Fig. S2, S3). This extended seawater isotope dataset contains 1550 datapoints from the open-ocean, with the following fit and a root mean square error (RSME) of 3‰:

$$\delta^2H_{sw} = 6.58 \times \delta^{18}O_{sw} - 0.12 \quad (R^2 = 0.82) \tag{1}$$



We will refer to this correlation as modern open-ocean waterline (MOOWL). The relationship between the hydrogen isotopic composition of the seawater and the measured salinity in the modern open-ocean is based on 903 datapoints obtained from the extended seawater isotope dataset:

$$\delta\ ^2H_{sw} = 3.53 \times S - 122.57 \ (R^2 = 0.66) \tag{2}$$

Finally, the oxygen isotopic composition of the seawater and the salinity were correlated based on 5602 datapoints:

$$\delta^{18}O_{sw} = 0.52 \times S - 18.02 \ (R^2 = 0.64) \tag{3}$$

### 2.5 Calculation of Past Seawater Isotopes

#### 2.5.1 Bottom Seawater

The oxygen isotopic composition of foraminifera ($\delta^{18}O_{Foram}$) depends mainly on the temperature of calcification (T) and isotope ratio of the water ($\delta^{18}O_{SW}$). McCrea (1950) published the first laboratory oxygen-isotope paleotemperature equation which

was subsequently revised and extended to different calcite sources and foraminifera species (e.g. Bemis et al., 1998; Cramer et al., 2011; Epstein et al., 1953; Lynch-Stieglitz et al., 1999; Mulitza et al., 2003; O'Neil et al., 1969; Shackleton, 1974; Spero et al., 2003). The relationship can be described as the following, where d is the slope and c is the intercept:

$$T = c + d \times (\delta^{18}O_{Foram} - \delta^{18}O_{sw}) \tag{4}$$

This relationship is used to reconstruct past $\delta^{18}O_{SW}$ values using temperature proxies and obtain salinity estimations (Lamy et

al., 2002; Pearson, 2012; Tang and Stott, 1993; Ganssen et al., 2011). The most relevant paleotemperature equation for benthic foraminifera is currently from Lynch-Stieglitz et al. (1999) adjusted to V-SMOW by Cramer et al. (2011). The $\delta^{18}O$-paleotemperature relationship is estimated with the in-situ benthic foraminifera from the genera *Cibicidoides* and *Planulina* (Lynch-Stieglitz et al., 1999). Recent studies adjust data of other genera to fit with those (e.g. Cramer et al., 2009, 2011). We adjusted the oxygen isotope ratios of *Uvigerina* in this study by 0.64‰ (i.e., Shackleton, 1974) for the $\delta^{18}O_{SW}$ reconstruction.

Important to note is that $\delta^{18}O_{SW}$ is always measured on the V-SMOW scale whereas calcite is standardized to V-PDB. The accepted value to convert V-PDB into V-SMOW at the time of Eq. (4) (Cramer et al., 2011) is 0.27‰.

$$T = 16.1 - 4.76 * (\delta^{18}O_{Foram} - (\delta^{18}O_{sw} - 0.27)) \tag{5}$$

Rearranged to $\delta^{18}O_{SW}$:

$$\delta^{18}O_{SW} = \frac{-16.1 + 4.76 \times \delta^{18}O_{Foram} + T}{4.76} + 0.27 \tag{6}$$






### 2.5.2. Surface Seawater

To estimate surface $\delta^2H_{SW}$ signal from the measured hydrogen isotopic composition of alkenones we used the suspended organic matter calibration (SPOM) from Gould et al. (2019):

$$\delta\ ^2H_{C37} = 1.48\ (\pm 0.4) \times\ \delta\ ^2H_{SW} - 199\ (\pm 3) \tag{7}$$

The environmental $\delta^2H_{C37}$ - $\delta^2H_{SW}$ calibration from Gould et al. (2019) is based on alkenones from open-ocean Group III haptophytes living in the Atlantic and Pacific ocean, covering large environmental differences such as salinity, temperature, nutrient concentrations and light intensity that in turn affect growth rate, for instance. We calculated a root-mean-square error (RMSE) of 5.8‰ for this calibration using the reduced data set of Gould et al. (2019).

An alternative environmental calibration is the sediment core-top study from Mitsunaga et al. (2022), where they extended and
revised the data of Weiss et al. (2019a):

$$\delta\ ^2H_{C37} = 1.44\ (\pm 0.13) \times\ \delta\ ^2H_{SW} - 191.62\ (\pm 1.13) \tag{8}$$

Mitsunaga et al. (2022) reports an RSME of 7‰. We estimated the error in the glacial-interglacial change in hydrogen isotopic
composition of the surface water ($\Delta\delta^2H_{SW}$) based on the error of the slope of the equation (Table 1), since we are interested in reconstructing changes rather than absolute values, with the assumption that the environmental relationship did not change over time.

### 3 Results and Discussion

### 3.1 Reconstruction of bottom seawater isotopes based on foraminifera

The oxygen isotopes of benthic *Uvigerina* in ODP core 1235 ranges between 4.15‰ to 2.27‰ (Fig. 1). Between 40 to 27 ka the $\delta^{18}O$ is relatively stable between 3.13 to 3.96‰, followed by a decrease towards 2.27‰ at 25 ka. Between 25 and 20 ka, the $\delta^{18}O$ signal is more positive with values of 3.7-4.15‰ and rapidly shifts by 1.6‰ till values of 2.27‰ in the uppermost sample. This record is consistent with ODP core 1234, showing an LGM to 1 ka shift of ca. 1.7‰ in the $\delta^{18}O$ values of *Uvigerina* (de Bar et al., 2018).

To reconstruct $\delta^{18}O_{SW}$, we need to reconstruct bottom water temperatures which unfortunately we were not able to do. To obtain some indications of temperature change we used published SST records from the same core. The organic SST proxy $U^{K'}_{37}$ measured in this core (from Varma et al., *under revision*) indicates a temperature range of 11.9 °C to 16.9 °C, with relatively stable temperatures between 19 and 40 ka of ca. 13 °C and an increase of ca. 4 °C until 7 ka, after which it remained relatively stable (Fig. 1). The TEX$^H_{86}$ (from Varma et al., *under revision*) reveals temperatures between 10.3 and 15.9 °C and
a similar increasing trend of 5 °C from 20 ka till ca. 1 ka. Thus, both proxies indicate a LGM to recent temperature change of





ca. 4 °C, consistent with SST proxy records from ODP core 1234 (de Bar et al., 2018). Adkins et al. (2002, 2001) suggested a similar temperature change of 4 °C for the Pacific, Southern, and Atlantic deep ocean and thus we assume a similar temperature change for the bottom water as reconstructed for the surface. Combined with the $\delta^{18}O$ values of *Uvigerina* this translate to a $\delta^{18}O_{SW}$ shift of 0.8 ±0.2‰ for ODP 1235 and 1234 from LGM to 1 ka using Eq. (6). This $\delta^{18}O_{SW}$ shift subsequently translates,

according to the modern open-ocean $\delta^{18}O_{SW} - \delta^2H_{SW}$ relationship described in Eq. (1), to an $\delta^2H_{SW}$ shift of ca. 5 ±4‰ for bottom waters (Table 1). This is similar to the values reported by Schrag et al. (2002) who measured oxygen and hydrogen isotopes of pore fluid from deep-sea sediments in the Southern and Northern Atlantic Ocean and observed a shift in $\Delta\delta^{18}O_{SW}$ of 0.7-1.1‰ and $\Delta\delta^2H_{SW}$ of 6-9‰ during the last deglaciation.

### 3.2 Reconstruction of surface seawater isotopes based on alkenones

The $\delta^2H_{C37}$ alkenone in core ODP 1235 range between -193‰ to -169‰ (Fig. 2). Between 40 ka and 25 ka the stable isotope ratio is relatively stable at ca. -180‰, then increases to -169‰ and decreases again to -173‰ at the LGM (ca. 20 ka) and down to the more negative value of -193‰ in the most recent sample. This record compares well, both in trend as well as in absolute values with that of ODP 1234 (Weiss et al., 2019a; Fig. 2) confirming that the trends in $\delta^2H_{C37}$ happened on a regional scale. If we calculate the overall shift from LGM (21 ka) to the most recent sample, then both records show a shift in $\delta^2H_{C37}$ of ca. -

20‰. Published $\delta^2H_{C37}$ alkenone records from Late Quaternary sediment cores (Kasper et al., 2014; Pahnke et al., 2007; Petrick et al., 2015; Simon et al., 2015 and Weiss et al., 2019b) report $\delta^2H_{C37}$ shifts which are very similar in magnitude (see overview in Table 1) and average 22±5‰ for the last deglaciation, indicating that this shift in hydrogen isotopic composition of $C_{37}$ alkenones is globally surprisingly similar in magnitude, despite the large differences in oceanographic settings.

Since Mix et al. (2003) observed the dominance of the open-ocean species *E. huxleyi* coccoliths in ODP 1234 and Hagino et

al. (2004, 2006) and Menschel et al. (2016) showed that the dominant alkenone producer on the west coast of South America is the Group III haptophyte *E. huxleyi*, it is most likely that alkenones in both cores reflect an open-ocean Group III haptophyte signal. We can therefore apply the SPOM calibration of Gould et al. (2019) (Eq. 7) to estimate the hydrogen isotopic composition of the seawater from the $\delta^2H_{C37}$. The resulting record (Fig. 2) shows a $\delta^2H_{SW}$ value of ca. 4 ±0.2‰ for the uppermost sample and ca. 18 ±0.7‰ for the LGM for ODP 1235 (Table 1). This indicates a shift of ca. 14 ±0.8‰ in $\delta^2H_{SW}$

from LGM to the most recent sample. Application of the core-top calibration from Mitsunaga et al. (2022) (Eq. 8) gives an identical change of 14 ±2‰.




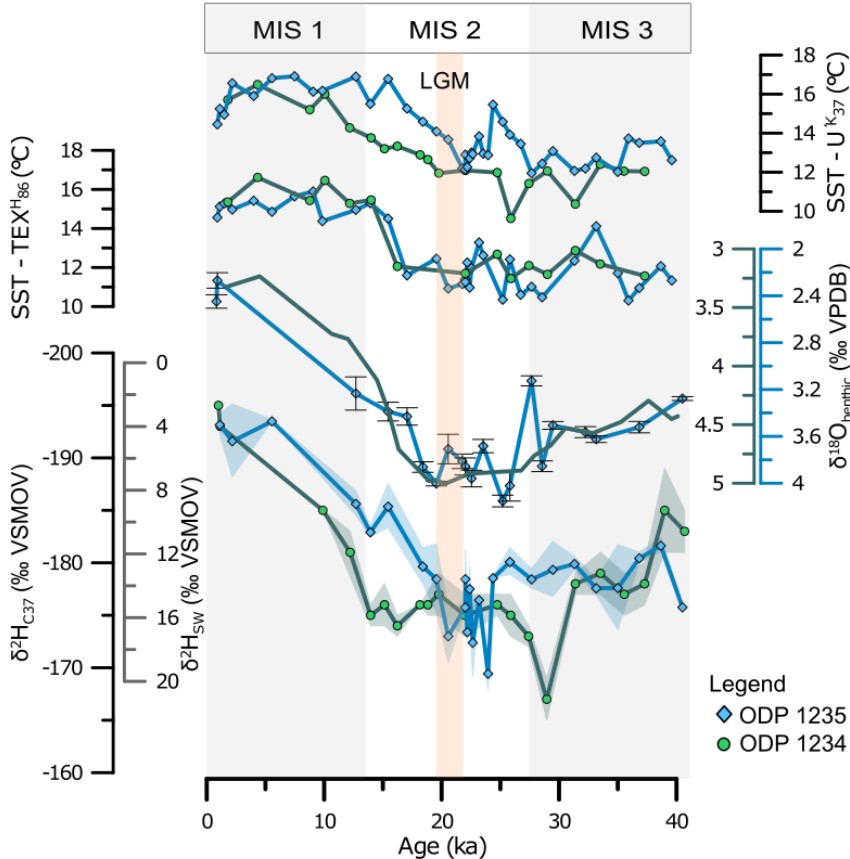

**Figure 2:** Hydrogen isotope ratios of ODP 1235, blue diamonds and ODP 1234 (Weiss et al., 2019), green circles, translated into $\delta^2$H of sea water ratios with Gould et al., 2019, shading represents the standard deviation of duplicate isotope measurements. Oxygen isotopes of benthic foraminifera and organic proxy temperature reconstruction of ODP core 1234 and 1235. $\delta^{18}$O of *Uvigerina* from ODP 1235 in blue with diamonds, error bars indicate standard deviation, 5pt average $\delta^{18}$O of benthic Foraminifera from ODP 1234 in black with circles. Sea water temperature data from ODP 1234 are from de Bar et al. (2019), and from ODP 1235 from Varma et al. (*under revision*).

### 3.3 Comparison of seawater isotopes and paleo salinity implications

The reconstructed $\Delta\delta^2$H$_{SW}$ of ca. 14‰ from C37 alkenones is higher than that inferred from the benthic foraminifera isotopes (5 ±4‰) and measured in pore fluids (Schrag et al., 2002), implying that the isotopic composition of the sea surface water experienced a larger change during the last deglaciation compared to the bottom water composition at the Chilean Margin. This larger regional change could be due to changes in local climate resulting in increased precipitation and/or decreased evaporation or increased freshwater runoff, for instance. Both cores, ODP 1235 and ODP 1234, are located within the Chile-Peru upwelling system and near to two major river mouths Río Biobío and Río Itata, both draining large basins (Muratli et al., 2010a). However, de Bar et al. (2018) observed no major change in the contribution of terrestrially derived biomarker concentrations and in proxies for river transported material like the BIT index (Hopmans et al., 2016) and the C32 1,15-diol index (de Bar et al., 2016, Lattaud et al., 2017a, b).



To reconstruct salinity change we applied the modern open-ocean $\delta^2H_{SW}$ – salinity relationship (Eq. 2) to the reconstructed
$\Delta\delta^2H_{SW}$. The bottom water $\Delta\delta^2H_{SW}$, estimated from benthic foraminifera, translates to a change of 1.4 ±0.2 psu which is in line
with the $\Delta\delta^{18}O_{SW}$ to salinity translation (1 ±0.2 psu; Eq. 3) and with earlier estimations based on oxygen isotopes and sea level
reconstructions (e.g. Adkins et al., 2002; Duplessy et al., 2002; Oba et al., 2004; Waelbroeck et al., 2002). The sea surface
$\Delta\delta^2H_{SW}$ of ca. 14‰ based on $\delta^2H_{C37}$ translates to a regional change of ca. 3 ±1 psu, a larger reconstructed change in salinity
than for bottom waters, but similar to the estimate of Lamy et al. (2002) who reconstructed a sea surface paleosalinity shift
from 36 to 33 psu during the last 8 ka based on $\delta^{18}O$ of planktic foraminifera for the southern Chilean Margin core ODP 1233.
The observed freshening in the surface waters relative to bottom waters may be due to a stronger input in relatively low salinity
waters from the Southern Ocean (ACC/PCC) and/or more transport of low salinity Chilean Fjord Water (CFW) that, as a result
of higher temperatures and intensifying continental meltwater input, resulted in even lower salinity surface water at the study
site.

### 3.4 Global distribution of hydrogen isotope changes during last deglaciation

If we apply the same methodology as used for ODP sites 1234 and 1235 to other published alkenone hydrogen isotope records,
we calculate a remarkable similar shift from LGM to Recent in the $\delta^2H_{SW}$ of between 11-17‰ for very different parts of the
globe (Table 1), suggesting there might be a global aspect to this relatively large shift in sea surface hydrogen isotope
composition, compared to the smaller shift observed in bottom waters waters (e.g. Schrag et al., 2002). There could be several
reasons for this globally observed similar change in surface water hydrogen isotopic composition. For example, increasing
average global temperatures during the deglaciation could have intensified the hydrological cycle leading to high precipitation
rates and runoff (including meltwater) in many places across the globe, at least around South America and South Africa where
most records originate from. This increased freshening of surface waters may have led to an additional depletion in $^2H$
compared to those of bottom waters. Furthermore, the salinity-$^2H$ relationship in surface waters might have been different
during the glacial compared to the interglacial due to a different evaporation-precipitation balance. Rohling et al., (2007)
demonstrated the influence of a dominating evaporation on the slope of the isotope-salinity relationship. Indeed, Putman et al.
(2019) suggested that hydroclimate processes are important controls on the slope and intercept of local meteoric water lines.
Surface waters may be more sensitive for changes in this salinity-hydrogen isotope relationship due to the direct atmosphere
connection. Potentially there was a steeper salinity-$^2H$ slope of Equation (2) for surface waters, due to e.g. increased surface
seawater evaporation during the LGM. This slope may have been changing during deglaciation towards the present day value
and thus result in a large change in surface water $^2H$ despite a similar shift in salinity as that in bottom waters.

To verify these assumptions, further application of $\delta^2H_{C37}$ analyses of sediment cores at different open-ocean locations and
combined $\delta^{18}O$ foraminifera analyses or pore fluid oxygen and hydrogen isotope reconstructions (Schrag et al., 2002) are
needed to study the past meteoric open-ocean waterline of surface and bottom waters. Extending our understanding of the
(past) relationship of seawater isotopes and salinity could help to understand and reconstruct ocean current and ocean mixing
processes.



**Table 1:** Hydrogen isotope ratios of long-chain alkenones ($\delta^2H_{C37}$) for the most recent samples and for the Last Glacial Maximum (LGM ca. 21 ka), ODP 1234 data from Weiss et al., 2019. Calculated hydrogen isotopic composition of seawater with SPOM calibration (Gould et al., 2019) and sediment core top calibration (Mitsunaga et al., 2022). The error of $\Delta\delta^2H_{C37}$ is estimated with the error of the calibration slope and std. Oxygen isotope ratios of benthic foraminifera from ODP 1234 (see de Bar et al., 2019), LR04 stack (Lisiecki and Raymo, 2005). Oxygen isotopes of the seawater calculated with Cramer et al., 2011 and a temperature change estimate of 4 °C. For Petrick et al. (2015) a bottom water temperature of 2 °C is estimated. The error between $\Delta\delta^{18}O_{SW}$ translated to $\Delta\delta^2H_{SW}$ is constrained with the RMSE of the Equation (3).

| Paper | Modern | | | | LGM | | | | Difference | | |
|---|---|---|---|---|---|---|---|---|---|---|---|
| | Age (ka) | $\delta^2H_{C37}$ ‰ vs. VSMOW | SPOM cal. $\delta^2H_{SW}$ ‰ vs. VSMOW | Core top cal. $\delta^2H_{SW}$ ‰ vs. VSMOW | Age (ka) | $\delta^2H_{C37}$ ‰ vs. VSMOW | SPOM cal. $\delta^2H_{SW}$ ‰ vs. VSMOW | Core top cal. $\delta^2H_{SW}$ ‰ vs. VSMOW | $\Delta\delta^2H_{C37}$ ‰ vs. VSMOW | SPOM cal. $\Delta\delta^2H_{SW}$ ‰ vs. VSMOW | Core top cal. $\Delta\delta^2H_{SW}$ ‰ vs. VSMOW |
| Pahnke et al. (2007) | 0.3 | -221 | -15 | -20 | 21.1 | -198 | 1 | -4 | 23 | 16 ±0.6 | 16 ±3 |
| Kasper et al. (2014) | 3.5 | -192 | 5 | 0 | 21.1 | -173 | 18 | 13 | 19 | 13 ±0.9 | 13 ±2 |
| Kasper et al. (2015) | 0.8 | -189 | 7 | 2 | 21.3 | -196 | 2 | -3 | 7 | 5 ±0.3 | 6 ±1 |
| Petrick et al. (2015) | 2.7 | -205 | -4 | -9 | 23.1 | -180 | 13 | 8 | 25 | 17 ±0.6 | 17 ±2 |
| Simon et al. (2015) | 2 | -195 | 3 | -2 | 20.1 | -179 | 14 | 9 | 16 | 11 ±0.6 | 11 ±1 |
| ODP 1234 | 1 | -195 | 3 | -2 | 21.98 | -175 | 16 | 12 | 20 | 14 ±0.7 | 14 ±2 |
| ODP 1235 | 1.1 | -193 | 4 | -1 | 20.56 | -173 | 18 | 13 | 20 | 14 ±0.8 | 14 ±2 |

| | Age (ka) | $\delta^{18}O_{Foram}$ ‰ vs. VPDB | | | Age (ka) | $\delta^{18}O_{Foram}$ ‰ vs. VPDB | | | $\Delta\delta^{18}O_{Foram}$ ‰ vs. VPDB | $\Delta\delta^{18}O_{SW}$ ‰ vs. VSMOW | $\Delta\delta^2H_{SW}$ ‰ vs. VSMOW |
|---|---|---|---|---|---|---|---|---|---|---|---|
| ODP 1234 | 0.78 | 3.3 | | | 20.31 | 5 | | | 1.7 | 0.9 | 5.5 ±3 |
| ODP 1235 | 0.9 | 2.3 | | | 20.56 | 3.8 | | | 1.5 | 0.7 | 4.2 ±3 |
| Petrick et al. (2015) | 1.6 | 2.7 | | | 21.1 | 4 | | | 1.3 | 0.9 | 5.8 ±3 |
| LR04 stack | 0 | 3.2 | | | 20 | 5 | | | 1.8 | 1 | 6.5 ±3 |

*Note.* The differences in $\delta^2H_{C37}$ ratios between LGM and the modern in all records suggest a large $\delta^2H_{SW}$ shift. Except for the Mozambique Channel record (Kasper et al., 2015) which does not record a depletion between the LGM and modern (highlighted in gray) and therefore does not record the glacial interglacial shift but local changes of freshwater input.

## 4 Conclusion

The ODP 1235 sediment record shows a similar large shift in $\delta^2H_{C37}$ during the last deglaciation at the Chilean Margin as observed by Weiss et al. (2019) for nearby core site ODP 1234, demonstrating that the shift is regionally consistent. The hydrogen isotope ratios of alkenones are thus a reproducible paleo proxy for relative changes in $\delta^2H_{SW}$ and likely suggests changes in the salinity of the surface water. The reconstructed change in hydrogen isotopes of sea water is ca. 14‰, independent of calibrations. In contrast, the change in hydrogen isotopic composition of the bottom water reconstructed with the oxygen isotope signature of benthic foraminifera was only $\Delta\delta^2H_{SW} = 5 \pm 1$‰. This indicates a larger change of the salt content in the surface water than the bottom water. Globally distributed C37 alkenone records also suggest a larger $\delta^2H_{SW}$ change for the surface waters then bottom water isotopes reconstructed from benthic foraminifera and suggest globally more freshening of surface waters compared to bottom waters. Further application of $\delta^2H_{C37}$ analyses combined with $\delta^{18}O$ foraminifera analyses or pore fluid oxygen and hydrogen isotope reconstructions may improve our picture of the past ocean seawater isotopic composition and salinity changes between surface and bottom water.

**Data Availability**

This study contains supplementary material. Supplement file 1 includes the age model and additional figures of the modern open-ocean salinity and isotope dataset used and cited in this study. Supplement file 2 contains the analysed isotope data of 305 ODP 1235. The age model, sea surface temperature and isotope dataset are archived at PANGAEA (Hättig et al., 2023a,b; Varma et al., 2023).

**Sample availability**

All processed sediment samples are stored at NIOZ, i.e. TLE, Apolar, Ketone, Polar fractions of ODP 1234 and ODP 1235 core and sieved material of ODP 1235.

**Author contribution**

All four (co-) authors collectively contributed to the conceptualisation of this study. Devika Varma and Katrin Hättig ordered samples from IODP and made sample splits for different analysis. Devika Varma prepared apolar, ketone and polar fractions and analysed and integrated alkenone and GDGTs for sea surface temperature. Katrin Hättig analysed the hydrogen isotopic composition of alkenones and performed the inorganic sample preparation, analysis, and age model computation. 315 Visualisation of research results and original draft preparation was done by Katrin Hättig. Stefan Schouten, Marcel T. J. van der Meer and Devika Varma reviewed and edited the original draft.

**Competing interests**

The authors declare that they have no conflict of interest.

**Acknowledgement**

We thank Michelle Penkrot from IODP Gulf Coast Repository for providing samples of ODP site 1235. Jort Ossebaar and Ronald van Bommel are thanked for analytical support. This work was carried out under the umbrella of the Netherlands Earth System Science Centre (NESSC). This project has received funding from the European Union's Horizon 2020 research and innovation programme under the Marie Sklodowska-Curie, grant agreement No 847504.




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
