# Peer review of "Glacial-interglacial seawater isotope change near the Chilean Margin as reflected by $\delta^2\text{H}$ values of C37 alkenones."

_EGUsphere, 2023_

## Author Comment (AC1)

**Glacial-interglacial sea water isotope change near the Chilean Margin as reflected by δ²H of C₃₇ alkenones.**

Katrin Hättig[1], Devika Varma[1], Stefan Schouten[1,2], Marcel T. J. van der Meer[1]

[1] Department of Marine Microbiology and Biogeochemistry, NIOZ Royal Netherlands Institute for Sea Research, Texel, the Netherlands.

[2] Department of Earth Sciences, Faculty of Geosciences, Utrecht University, Utrecht, the Netherlands.

*Correspondence to*: Katrin Hättig (katrin.haettig@nioz.nl)

**Authors response:** *We thank both reviewers for their constructive comments on the manuscript, which will help to improve it. Please find a detailed reply to all comments below.*

**Reviewer 1:**

**General Comments**

**RC1:** In this paper, Hättig et al. generate a paired surface-benthic ocean temperature / hydrogen / oxygen isotope record from the Chilean Margin spanning the last 40 kyr. Their new data are similar to those from a neighboring site and they also show that the mismatch in planktic-benthic seawater isotope change is consistent to that observed in the handful of other extant LGM-modern alkenone hydrogen isotope records. This multi-proxy approach is creative and demonstrates the potential of the alkenone δ2H proxy. This manuscript also shows a good awareness of prior work and current technical and analytical limitations. I believe it is well within the purview of Climate of the Past and therefore recommend it for publication following extensive but relatively minor revisions.

My first suggestion is a careful grammatical overhaul; I am sure I did not identify every single instance where the language and logical flow could be improved and I would encourage the authors to actively seek out opportunities to do to so rather than merely addressing those listed below.

- **Authors:** *We thank the reviewer for the technical corrections pointed out and we will take the opportunity to go through the text carefully.*

My other main recommendation would be a more extensive treatment of the uncertainties. This work involves several transfer functions (i.e., δ18O to δ2H, δ18O to salinity, foraminiferal δ18O to seawater δ18O, alkenone δ2H to seawater δ2H), calibration error, and general unknowns (benthic paleotemperatures); the authors do a good job of bringing these to our attention, but I believe a slightly more quantitative assessment of how these compounded uncertainties (roughly calculated) affect their final conclusions would be beneficial. (E.g., does this mean that the last deglaciation is the smallest change we can reasonably do this type of analysis for? Is the surface-depth isotope offset within error?)

- **Authors:** *We will try to draw and include statements on how these uncertainties affect the final conclusions. There are three main components one needs to consider: 1) The reproducibility and accuracy error of compound-specific hydrogen isotope analysis which is ca. 5‰ based on replicate analysis for the standards (for MIX B the source is A. Schimmelmann, Indiana University). 2) The calibration error between $\delta^2H_{SW}$ and $\delta^2H_{C37}$ reflected by the root-mean-square error of 5.8‰ for the calibration from Gould et al. (2019) and 7‰ for Mitsunaga et al. (2022). 3) The variability of the modern open-ocean waterline with an RMSE of 3‰. All three components result in a propagated error of ca. 9‰ for the absolute $\delta^2H_{SW}$ estimations. However, the error of the relative change in $\delta^2H_{SW}$ is likely smaller. For example, when we use the error of the calibration slopes (Gould et al., 2019; Mitsunaga et al., 2022, which is ca. 3‰, and of the water line, which is ca. 1‰), then the propagated error of $\Delta\delta^2H_{SW}$ is approximately 6‰.*

**Specific Comments**

**RC1:** Lines 52-56: My understanding was that the alkenone-seawater hydrogen isotope fractionation's salinity-dependence was observable in culture (Schouten 2006 and your other references) but not always in sediment (Weiss et al., 2019; Mitsunaga et al., 2022). I think you can say this and acknowledge that effect of salinity on alkenone-water fractionation is not settled.

- **Authors:** *Yes, the impact of salinity is unclear, we will rephrase these lines.*

**RC1:** Line 73: I think you should put a sentence around here on why you are using Site 1235; explicitly call out the benefits of having the nearby Site 1234 δ2HC37 and temperature records, then get into why your site and approach are different (which you do: different depths, etc.). Also, in general, especially in your abstract and introduction, you can do a better job saying why your study is so important and why you needed to make these measurements so close to an existing δ2HC37 A sentence or two here or there could make a big difference.

- **Authors:** *We will add text at the different parts of the revised manuscript making it clearer why we chose site 1235 and how it is beneficial.*

**RC1:** Lines 94-97: Were planktic foraminifera unavailable? That could be another means of reconstructing surface water δ18O. And then you could avoid the δ18O- δ2H transfer function, which is an additional source of error…

- **Authors:** *Planktic foraminifera were picked when abundant in core slides and their $\delta^{18}O$ was analysed. However, this record did not cover the whole 20 ka shift sufficiently enough to calculate the surface oxygen isotope water shift during the last deglaciation.*

**RC1:** Lines 128-130: How different were the δ2HC37:3 and δ2HC37:2 values? I.e., was there any C37:3-C37:2 (inter-alkenone) fractionation? I think you should mention it in your results; I know it is not the main thrust of this paper, but some people (Sachs, D'Andrea) have observed an offset while others (Weiss, Mitsunaga) have not, and this could be of interest to the biochemists studying coccolithophore cell water dynamics. At the very least, if there is no (statistically significant) offset, mentioning that legitimizes your choice to report combined δ2HC37.

- **Authors:** *Yes, to account for inter-alkenone fractionation we integrated $\delta^2H_{C37:3}$, $\delta^2H_{C37:2}$ separately as well as combined peaks of $C_{37:2}+C_{37:3}$ alkenones, $\delta^2H_{C37}$. Our analyses allowed us to do so in one measurement. All values are reported in the data file on Pangaea (https://doi.org/10.1594/PANGAEA.958880). The submitted supplement file reports only the integrated $\delta^2H_{C37}$ and $\delta^2H_{C38}$ values, we will add the individual $C_{37}$ alkenone values to the supplement. The $C_{37:3}$ alkenone suggests a $\delta^2H$ shift of 24‰ and the $C_{37:2}$ alkenone a shift of 16‰. However, van der Meer et al. (2013) have shown that a weighted average or combined integration of $C_{37}$ peaks might be more appropriate than individual values when using alkenone hydrogen isotopes for reconstructing paleo sea surface salinity changes. An additional reason for using the combined $C_{37}$ alkenone values is that older datasets are all based on the integrated $C_{37}$ alkenones and we compare our results to these datasets (e.g. Pahnke et al., 2007; Kasper et al., 2014). Lastly, the published $\delta^2H_{C37}$ - $\delta^2H_{SW}$ calibrations (Mitsunaga et al., 2022; Gould et al., 2019) are also based on the combined $C_{37}$ peak), we therefore prefer to stick to this.*

**RC1:** Lines 137-150: I think an obvious question is why bother with converting δ2HC37 to δ2HSW to salinity if there are existing δ2HC37-salinity calibrations, and I think you need to address this, at least briefly. What could be said is that the direct salinity calibrations are worse than the δ2HC37 to δ2HSW ones in core-tops (Mitsunaga, Weiss). Of course, then you have to explain why δ2HC37-salinity is worse than δ2HC37-δ2HSW if the δ2HSW-salinity relationship is so tight...

- **Authors:** *As we mentioned in the Introduction at line 55-69, culture $\delta^2H_{C37}$-salinity calibrations are based on single species and fixed conditions, which could be problematic when translating to environmental calibrations. The core-top calibration should be the most suitable since it includes multiple species and environmental variability including sedimentation and to some degree preservation. However, salinity changes become very large when applying these $\delta^2H$-salinity relationships, suggesting that we are missing a piece of the puzzle (discussed in Weiss et al., 2019b). The SPM calibration of $\delta^2H_{C37}$ to $\delta^2H_{SW}$ also includes contributions from different species and the effects of other environmental parameters and fits very well with the updated core-top calibration of Mitsunaga et al. (2022). With the reconstructed $\delta^2H_{SW}$ information, salinity interpretations can carefully be made with a global (as demonstrated) and local waterline.*

**RC1:** Lines 195-208: I get that you were unable to measure benthic paleotemperatures, but that does introduce some holes in your argument. It is good that you have the Schrag et al. bottom-water temperatures, because otherwise I would be skeptical that planktic and benthic warming would be the same. Is there any way you could show how much the uncertainty in your calculations matters? I.e., if you are off by X °C, how much does it matter to your calculations of benthic $\delta^{18}O_{SW}$ change? And then how does that error propagate to $\delta^2H_{SW}$ calculations?

- **Authors:** *We agree with both reviewers that the bottom water isotope reconstruction with respect to the temperature uncertainty needs more attention. We therefore performed a sensitivity analysis (see below) which showed that one degree of temperature change affects the $\delta^{18}O_{SW}$ shift by 0.2‰ and the $\delta^2H_{SW}$ by 1.4‰. In fact, the sensitivity analysis shows that even with negligible bottom water temperature change, the magnitude of $\delta^2H_{SW}$ change of bottom waters will still be lower than that of surface waters ($\delta^2H_{SW} = 14‰$).*

  *The following table shows the impact of different bottom water temperature changes ($\Delta T$) on the bottom water oxygen and hydrogen isotope change at ODP Site 1234 and 1235. $\Delta\delta^{18}O_{foram}$ estimate is based on the average observed at sites 1234 and 1235 (see Table 1 in preprint).*

| $\Delta\delta^{18}O_{foram}$ | $\Delta T$ | $\Delta\delta^{18}O_{SW}$ | $\Delta\delta^2H_{SW}$ |
|---|---|---|---|
| 1.6 ±0.2‰ | 0 °C | 1.6 ±0.2‰ | 10.6 ±3‰ |
| 1.6 ±0.2‰ | 1 °C | 1.4 ±0.2‰ | 9.3 ±3‰ |
| 1.6 ±0.2‰ | 2 °C | 1.2 ±0.2‰ | 7.9 ±3‰ |
| 1.6 ±0.2‰ | 3 °C | 1 ±0.2‰ | 6.5 ±3‰ |
| 1.6 ±0.2‰ | 4 °C | 0.8 ±0.2‰ | 5.1 ±3‰ |

**RC1:** Lines 249-259: Makes sense – if you assume the same LGM-modern temperature change, you get similar changes in benthic $\delta^{18}O$ as global. If you talk about temperature uncertainties earlier, this might be a good time to discuss what you discovered. What if deglacial benthic temperature change (and therefore $\delta^2H_{SW}$ change) was more / less than expected based on Schrag et al. numbers? Does it come close to affecting your argument that planktic $\delta^2H_{SW}$ change was much greater than benthic?

- **Authors:** *If the bottom water temperature change was higher than the assumed 4 °C, this would result in a smaller seawater isotope change and therefore smaller bottom water salinity change. Even if the bottom water temperature change is close to 0, then the seawater isotope change of the bottom waters is still smaller than that of surface waters.*

**RC1:** Line 261-281: I think you need to qualify this a bit. It is a very interesting trend if true, but since there are only seven records—including the anomalous Mozambique channel Kasper et al. one—from mostly low to mid-latitudes, as you acknowledge, I am not sure the data is there to do so quite yet. Presumably, there are other Mozambique channel-type sites out there, we just have not discovered them yet. I think you need a sentence or two at the start of this section acknowledging the small number of extant records. But as long as the trend is real...

- **Authors:** *We agree that the published records do not cover the entire globe and definitely not equally, though we note that it will be some effort to generate substantially more records. We will take this into account in our discussions.*

**RC1:** Lines 268-269: I think it is believable that the surface ocean would have freshened first, but I think it is intriguing that these changes would not have propagated to the deep ocean within several thousand years given the mean mixing time of seawater. Do you have any other sources that could support this?

- **Authors:** *We are not aware of any other studies which could support this. We agree to the reviewer that due to mixing over thousands of years we would expect a propagation of the surface changes eventually towards the bottom. However, modern hydrogen isotope data of the ocean is limited and therefore the distribution and underlying processes in the surface water versus the bottom water are not well known yet.*

**RC1:** Lines 274-275. I think you could come right out and say that the LGM was drier than the modern. A source or three here could really strengthen your argument and / or I think you need to flesh out the mechanisms by which a drier atmosphere could change the salinity- $\delta 2H$ relationship, which then more directly affects the surface than the depths. (That last part makes perfect sense to me.)

- **Authors:** *We thank the reviewer for this suggestion and we will look for sources to strengthen the argumentation.*

**RC1:** Table 1: I think you need to explain in the text or table caption somewhere why the Mozambique channel Kasper record is excluded from your calculations (i.e., changing riverine inputs rather than seawater $\delta 2H$ change). I would also use "source" instead of "paper" for the leftmost column.

- **Authors:** *Thank you for the suggestion, we will make it clearer why the Mozambique channel data is excluded (e.g., high BIT index, changing distance to the river mouth). We will change "paper" to "source".*

**Technical Corrections**

**RC1:** Line 2: [Here, and subsequently] I do not have strong feelings about this, but I have been told that delta notation—i.e., "δ2H" or "δ18O"—is an adjective and therefore that it always needs to be followed by a noun, i.e., "δ2HC37 values," "this δ2HC37 record," "our new δ2HC37 data," etc.

- **Authors:** *We will check and fix this.*

**RC1:** Line 2: [Here, and subsequently] I believe that the number of carbons should be subscripted, i.e., "C37."

- **Authors:** *We will change this.*

**RC1:** Line 12: "Shift" is overused throughout the abstract, try to find synonyms.

- **Authors:** *We will improve this.*

**RC1:** Line 12: The last / most recent deglaciation?

- **Authors:** *Yes, we will adapt this.*

RC1: Line 16: Separate "waterline" into two words.

- **Authors:** *We will change this.*

**RC1:** Line 21: "Physiochemical"

- **Authors:** *We do mean to say physicochemical, as it refers to both physical and chemical attributes e.g., temperature, salinity, pressure, density.*

**RC1:** Line 26: "Local climate regime" is vague; do you mean temperature?

- **Authors:** *We mean different climate parameters such as precipitation, temperature, humidity.*

**RC1:** Line 28: "Low oxygen and hydrogen isotopic values" is vague, could be referring to D/H ratios, etc. If you mean "δ2H or δ18O values," just say so. (You do this in lines 30-31 anyway.)

- **Authors:** *Yes, we mean $\delta^{18}O$ and $\delta^{2}H$ values.*

**RC1:** Line 36: The "f" in "δ18Oforam" is sometimes capitalized, sometimes not. I would go with lowercase, but just be consistent. You also re-abbreviate this (Line 95) elsewhere; you only need to once.

- **Authors:** *We will make it consistent.*

**RC1:** Line 36: [Here, and elsewhere?] You already introduced an abbreviation for the oxygen isotopic composition of seawater (δ18Osw), might as well use it.

- **Authors:** *We will change this.*

**RC1:** Lines 43-45: Also LGM porewater δ18Osw measurements; see Adkins & Schrag 2001.

- **Authors:** *We will add it here.*

**RC1:** Line 45: "To derive" should be "deriving."

- **Authors:** *We will change this.*

**RC1:** Line 61: What is the expected deglacial salinity change?

- **Authors:** *In line 43-44 we describe the oxygen isotope change; we should add that based on this data and modelling attempts the expected salinity change is 1-2 psu.*

**RC1:** Line 70: "Alkenone" should be "alkenones."

- **Authors:** *We will change this.*

**RC1:** Figure 1: The location of the word "Chile" is misplaced. I get not wanting to put it near where Chile actually is, because it's a bit crowded there, but maybe that means it is unnecessary. There are also several typos in the caption.

- **Authors:** *We will change this.*

**RC1:** Line 85: [Here, and elsewhere] I would mostly use "site" instead of "core" in this section since you are referring to the physical locations of the cores rather than the cores themselves. I would use "core" if you were discussing the characteristics of the core (i.e., length, lithology, etc.).

- **Authors:** *We thank the reviewer for this suggestion and will change it accordingly.*

**RC1:** Line 86: Remove "located."

- **Authors:** *We will remove it.*

**RC1:** Line 88: Add a comma between "upwelling" and "stimulating."

- **Authors:** *We will add a comma.*

**RC1:** Line 96: [Here, and elsewhere] be consistent with using a comma after "i.e." or "e.g." I believe the difference is British versus American English; see which the journal follows.

- **Authors:** *Indeed, the difference is American English and British English. We follow in the text British English and will therefore remove the commas.*

**RC1:** Lines 96-97: This is a run-on sentence.

- **Authors:** *We will try to improve this.*

**RC1:** Line 106: Misplaced parenthesis.

- **Authors:** *We will change this.*

**RC1:** Lines 107-108: "Ka" should be "kyr." (Ka [or Ma] specifically means years ago, where here I think you just mean x cm per x thousand years.) Also, you do not need the periods between "cm" and "kyr."

- **Authors:** *We will change this.*

**RC1:** Line 111: Check if you need the hyphens here. I am not sure that you do.

- **Authors:** *We will change the hyphens.*

**RC1:** Line 114: Is the standard deviation referring to NBS-19 or the internal standard or both?

- **Authors:** *It is referring to both. We always try to have both standards within 0.1‰. NBS19 is the official IAEA reference standard and is very homogeneous. The NFHS1 in house standard is also foraminifera material, but less homogeneous.*

**RC1:** Line 115: Both might be acceptable, but I have more commonly seen "per mille." Also, I do not think you need to define the ‰ symbol, but if you do, you should do so the first time you use it, not here.

- **Authors:** *We will change this.*

**RC1:** Line 133: [Here, and elsewhere] to clarify, "online" refers to the value measured in the lab that day and "offline" refers to the published / "correct" values (via Schimmelmann or others)?

- **Authors:** *The reviewer is right. Alternatively, we will replace this with predetermined values for in house standards and certified values for certified standards (in this case MIX B from A. Schimmelmann, Indiana University).*

**RC1:** Lines 134-135: [Here, and elsewhere] be consistent with spacing between mathematical symbols (±, <, ‰, etc.). Check journal-specific conventions.

- **Authors:** *OK. Between symbol and number no space e.g. -173, ±5, between number and ‰ no space as it is not a Unit. Between number and unit space.*

**RC1:** Line 144: "RSME" should be "RMSE."

- **Authors:** *We will change this.*

**RC1:** Line 145: "Waterline" should be two words.

- **Authors:** *We will change this.*

**RC1:** Line 162: Italicize "in situ," but I cannot tell what it adds here. (Do you mean from sediment versus from the water column or culture?) Try "estimated using the benthic genera Cibicidoidies and Planulina."

- **Authors:** *The equation by Lynch-Stieglitz et al. (1999) as rearranged by Cramer et al. (2011) is calculated from core-top samples, i.e. a field calibration. Pearson et al. (2012) described this calibration as* "in situ".

**RC1:** Line 163: Not sure these are recent. "Subsequent," maybe? Also, phrased a little awkwardly ("other species' values are projected onto the Cibicidoidies and Planulina function?").

- **Authors:** *We will change it to: "Uvigerina genera $\delta^{18}O$ values are corrected to Cibicidoidies and Planulina $\delta^{18}O$ values. "*

**RC1:** Lines 251-252: I am wondering if you could use separate abbreviations for surface and deep water δ2HSW values and change – δ2HSWp and δ2HSWb? I am sure you can think of something better and / or do not be afraid to use synonyms for "surface" and "deep" ("planktic" and "benthic," etc.) too.

- **Authors:** *We think it is good to stick to surface and bottom water, this is how the article from Results to Discussion is structured and we think this will be clear to the proxy and modelling community. We hesitate to use planktic and benthic as it could lead to more confusion, and we like the reader to focus on the water masses, rather than the proxies. We thank the reviewer for the suggestion, and we will use the abbreviation $\delta^2H_{SSW}$ for surface seawater and $\delta^2H_{BSW}$ for bottom seawater. This would be similar to the abbreviations used for sea surface and bottom temperature i.e. SST and BWT.*

**RC1:** Line 262: "Remarkable" should be "remarkably," although I would probably just take it out.

- **Authors:** *We will change this.*

**RC1:** Line 265: "Waters" typo.

- **Authors:** *We will change this.*

**RC1:** Line 269: "Salinity-δ2H" instead of "salinity-2H?"

- **Authors:** *We will change it to $\delta^2H$ to avoid confusion.*

**RC1:** Line 285: "Std" = standard deviation?

- **Authors:** *Yes, with Std we mean the standard deviation and will expand the abbreviation.*

---

## Author Comment (AC2)

**Glacial-interglacial sea water isotope change near the Chilean Margin as reflected by $\delta^2$H of C$_{37}$ alkenones.**

Katrin Hättig[1], Devika Varma[1], Stefan Schouten[1,2], Marcel T. J. van der Meer[1]

[1] Department of Marine Microbiology and Biogeochemistry, NIOZ Royal Netherlands Institute for Sea Research, Texel, the Netherlands.

[2] Department of Earth Sciences, Faculty of Geosciences, Utrecht University, Utrecht, the Netherlands.

*Correspondence to*: Katrin Hättig (katrin.haettig@nioz.nl)

**Authors response:** *We thank both reviewers for their constructive comments on the manuscript, which will help to improve it. Please find a detailed reply to all comments below.*

**Reviewer 2:**

**General Comments**

**RC2:** Hättig et al. present an interesting new collection of paired surface ocean $\delta^2$H$_{C37}$ and benthic $\delta^{18}$O$_{FORAM}$ records from the Chilean Margin for reconstructing changes in ocean isotopic composition ultimately speaking to water mass and/or salinity variability from the LGM.

The data show good agreement with a previously published record from the same locality, and the treatment of sample and data analyses are consistent with those accepted in the literature. The authors' new approach/application of hydrogen isotopic composition of alkenones for reconstructing the hydrogen isotopic composition of surface water (and then ultimately of salinity) is unique and interesting and certainly warrants publication. I believe this body of work is suited to Climate of the Past, following careful revision.

My overall recommendation to the authors would be to carefully consider reviewing the paper for clarity and detail. There are many instances of grammatical errors/sentences that are difficult to follow. In particular, the discussion and even more so the conclusion is not as well organized as it could be. The conclusion should be a thoughtful overview of the major findings of the work and as it is written it is relatively difficult to follow. I strongly encourage the authors to review the organization of this section and to also improve on the overall portrayal of the major conclusions from this work.

- **Authors:** *We thank the reviewer for the positive response. We will carefully go through the text, especially the conclusions, and improve the structure and language.*

Next, if the authors could be clearer and more explicit about the errors that they are reporting throughout, it would be a more powerful reflection of the data. There are of course assumptions that must be made throughout; however, I also wonder if the authors would consider reporting the error in the transfer functions used (i.e., Equations 1 through 8).

- **Authors:** *We agree with the reviewer that clear reporting of data and errors is important. We, therefore, display the dataset of the modern open-ocean seawater isotopes and salinity in figure S2 and S3 of the supplemental information. The root-mean-square errors (RMSE) for the transfer functions (1, 7, 8) give an indication of the possible error range for the reconstructed $\delta^2H_{sw}$ from $\delta^2H_{C37}$ and $\delta^2H_{sw}$ vs $\delta^{18}O_{sw}$ values in ‰VSMOW. We refer to our reply to reviewer's 1 comment where we specify the propagated error, which is ca. 9‰ for the absolute $\delta^2H_{SW}$ values and ca. 6‰ for $\Delta\delta^2H_{SW}$.*

**RC2:** Because much of the discussion and conclusions deal with the offset found between surface and benthic estimations/reconstructions of $\delta^2H_{sw}$, and because the reconstruction specifically of benthic $\delta^2H_{sw}$ relies on temperature, which the authors do not model, I wonder if the authors considered running (and then reporting on) an experiment where they vary the bottom temperature estimates for the benthic reconstruction of $\delta^2H_{sw}$ to determine a conceivable (appropriate) range (and to demonstrate the sensitivity) of this particular offset (Lines 195-208). It is difficult to follow that the authors assume temperature is consistent at depth when they report a major shift in isotopic/salinity signatures between surface and benthic environments. Whether this can be resolved here or not, I do believe the authors should spend some time in their discussion on this discrepancy. Are there no planktonic foraminifera $\delta^{18}O$ data to use for reconstructing surface salinity?

- **Authors:** *We thank the reviewer for this comment which is identical to a comment made by reviewer 1. We, therefore, refer to our reply to reviewer's 1 comment where we perform a sensitivity analysis which shows that even with unrealistically small bottom water temperature changes, the change in $\delta^2H_{sw}$ of bottom waters would still be lower than that of surface waters.*

**RC2:** Finally, I wonder why the authors did not test the $\delta^2H_{C37}$ – salinity calibration models for reconstructing salinity at this site (at the very least plotting their new $\delta^2H_{C37}$ data alongside previously published data). The authors mention this calibration in the literature; however, the paper overall might be made more powerful and rigorous by including this evaluation.

- **Authors:** *The $\delta^2H_{C37}$ –salinity calibrations based on culture studies (e.g., Schouten et al., 2006.), core-top studies (Weiss et al., 2019) and SPOM filters (Gould et al., 2019) have been compared and discussed for the ODP Core Site 1234 at the Chilean Margin in the study of Weiss et al. (2019b). Our hydrogen isotope data from site 1235 is very similar to the last deglaciation $\delta^2H_{C37}$ shift reported by Weiss et al. (2019b) and other global sites. Therefore, we did not see the need to repeat this discussion here, though we will briefly summarize it in a revised manuscript.*

**Specific Comments:**

**RC2:** Line 9/the overarching argument that the work presented illustrates a 'regional' not 'local' phenomena/record: I suppose it is relatively subjective, however, it is my experience that a regional phenomenon (especially in the vast ocean environment) would require the authors to have evaluated core samples taken from a much larger range of latitude/locations along the Chile margin. In my experience, 12km distance between two cores is showing consistency at one specific local, where the two cores would be treated essentially as replicate records to illustrate a robust finding. I suggest the authors reconsider whether their work presented is truly extending a local record to a regional one with just two cores so close in proximity. Although this does not detract from the importance of their record, nor does it negate the impact two core records has on improving the results.

- **Authors:** *The reviewer is correct that the distance between the two cores is only 12 kilometers, but the water depth from which the cores are retrieved is quite different and therefore still represents somewhat different conditions. Thus, although the cores are from very close proximity, we feel that "local" does not entirely do this study justice, but we agree with the reviewer that "regionally" is perhaps making it too big. However, we do note that a large number of palaeoceanographic studies draw regional conclusions by extrapolating results from a single core to a whole region. Here we extrapolate the results of two replicate records to a whole region which is, in our view, more robust than most other studies. Furthermore, the comparison to other records confirms that the signal is not reflecting local circumstances (e.g. river input) but regional and perhaps global changes.*

**RC2:** Throughout the manuscript, I would also suggest the authors be very clear and careful about describing isotopic trends/directionality. Whenever a change is indicated in isotopic composition through time, it would be very helpful to the reader to know in which direction the change occurred (i.e., the seawater has becoming enriched in $\delta^{18}O$ for example). This is an important detail that will aid the reader throughout.

- **Authors:** *We agree that this needs to be clearer.*

**RC2:** Figure 2 and Table 1: The figure and table headings here are a little bit difficult to follow. I suggest (as was done nicely in Figure 1) the authors start the headings with a more general/overarching statement of what the figure/table are describing/illustrating before describing specific aspects of the figure/table. Ensure that the figure/table titles can be read independently of the main text and be fully understood.

- **Authors:** *We agree with this and will change accordingly.*

**Technical Corrections:**

**RC2:** Line 2: subscript on C37 ($C_{37}$)

- **Authors:** *We will correct it.*

**RC2:** Line 9 and throughout the work: please determine whether you will spell the word 'seawater' as one word (as in my experience it is often spelled) or as two words 'sea water'. There are many instances of both iterations throughout the manuscript, and this should be cleaned up before publication.

- **Authors:** *We will make it consistent.*

**RC2:** Line 12: It is important I think to indicate that the second core reported on in this work is indeed a previously reported on/published record - this is good to indicate even in the abstract here and then also throughout.

- **Authors:** *We will make this clearer. However, both in the Abstract and in the Introduction we state that we present a new hydrogen isotope record of ODP core 1235. In the introduction line 76-77, as well as throughout the methods and results (e.g. line 210 and 213), we state clearly that we compare our record to published data of ODP site 1234 (i.e. de Bar et al., 2018 and Weiss et al., 2019b).*

**RC2:** Line 16: Make it clear here that "this suggests a shift of ca. 5 per mil in benthic/deep water".

- **Authors:** *We will make this clearer but, as stated in the reply to the comments of reviewer 1, we prefer to use the term bottom water, rather than deep or benthic.*

**RC2:** Line 21: again be clear about how you spell seawater – perhaps here the parentheses are not necessary and just spell out 'seawater'.

- **Authors:** *Thank you for pointing this out, we intend to spell it consistently as "seawater".*

**RC2:** Line 22: the phrasing 'which could improve' is a bit weak, perhaps just 'which improve our understanding'...

- **Authors:** *We will change it.*

**RC2:** Line 28: Be sure the use of the colon ":" throughout is necessary. For instance, this sentence could instead read: "... isotopes of seawater, where during glacial periods large amounts of water...."

- **Authors:** *We will correct it.*

**RC2:** Line 28: "Low oxygen and hydrogen isotope values" is not clear, do you mean more negative? It may be helpful to include/define the equations for $\delta^{18}O$ and $\delta^2H$ here. Since $\delta^2H$ especially is a relatively 'new' proxy for seawater salinity, defining the hydrogen isotopes explicitly could be informative.

**Authors:** *Yes, we mean more negative. We discuss the updated equations for $\delta^{18}O$ and $\delta^2H$ in the Material and Methods section.* **RC2:** Line 29: You might consider removing the word "the" in this sentence: "....ice on land, leaving seawater more enriched in heavier deuterium isotopes."

- **Authors:** *We will correct it.*

**RC2:** Line 30: Awkward wording choice to describe 'leading the formation of …'. Perhaps something along the lines of: "…the decrease in global ice volume releases low density fresh water with relatively low…" Also, consider throughout the manuscript when discussing isotopic values whether you are being consistent in your description of 'low vs high' or 'heavier vs light' or 'enriched' etc. isotopic signatures. Choosing one way to describe the directionality of the isotopes is much clearer.

- **Authors:** *Agree, we will use low and high values and highlight that low means more negative.*

**RC2:** Line 31: seawater

- **Authors:** *We will change it.*

**RC2:** Line 32: "… have a close relationship…" is perhaps not the most technical way to describe this, perhaps something more like "… are correlated with…" or "…are tightly coupled with…"

- **Authors:** *We will change it.*

**RC2:** Line 33/35: seawater (please check this throughout, I won't call it out in my further comments)

- **Authors:** *Thank you for pointing this out, we will check its consistent use.*

**RC2:** Line 36: I was always taught to refer to the "isotopic" composition of something… so perhaps "… is a function of the oxygen isotopic composition of ambient seawater…"

- **Authors:** *We agree and will check the text.*

**RC2:** Line 41: "… or clumped carbon isotopes ($\Delta_{47}$)…" for those unfamiliar with $\Delta_{47}$.

- **Authors:** *Thank you, we will change it.*

**RC2:** Line 41: "In some cases, independent temperature proxies based on organic…" The authors might consider including the word 'independent' here to indicate they are proxies not coming from forams themselves but other sources.

- **Authors:** *OK.*

**RC2:** Line 43: "…records and modelling…" is a bit awkward wording. Please consider re-evaluating this sentence structure.

- **Authors:** *We will try to improve the wording.*

**RC2:** Line 46/47: "… relationship between $\delta^{18}O_{sw}$ and salinity." The 'salt content' is strange wording.

- **Authors:** *We will change this.*

**RC2:** Line 47: Perhaps the last sentence would be clearer if written something like: "However, it is uncertain whether this relationship holds through time."

- **Authors**: *Thank you for the suggestion and we will address this in the updated manuscript.*

**RC2:** Line 50: Again, please be clear on the direction of the isotopic change.

- **Authors:** *We will indicate the direction of the shift clearer.*

**RC2:** Line 51: Perhaps define $C_{37:2}$ and $C_{37:3}$ here because you do not define this 'shorthand' earlier. Since you are opting to combine the signals, it is important to describe to the reader what these two parameters are.

- **Authors:** *We will indicate that $C_{37:2}$ and $C_{37:3}$ mean the long chain alkenone with 37 carbon atoms and two and three double bonds, respectively.*

**RC2:** Line 57: What $\delta^2H$ signature are you referring to here? The algal $\delta^2H$ response to changes in salinity? I believe here is where the authors describe the range in salinity reconstructions possible when using culture vs SPOM vs core top $\delta^2HC37$ – salinity calibration models. I do think it would be a powerful discussion to include a small discussion on what the data in this new core would show if a $\delta^2H_{C37}$-salinity calibration model was used to reconstruct salinity directly from $\delta^2H_{C37}$. Especially given the authors argue that the dominant species of alkenone producers in this region are likely to be *E. hux*

- **Authors:** *We refer to the alkenone $\delta^2H$, we may state this clearer. We agree with the reviewer's idea, however, we believe this was already extensively discussed by Weiss et al. (2019b). We will therefore refer the reader to that study and only briefly summarize it in the current manuscript (see response to the earlier comment on this topic)*

**RC2:** Line 60/61: This sentence is difficult to follow, larger than expected based on what?

- **Authors:** *We will rephrase this. We mean to say that the salinity changes Weiss et al. (2019b) reconstructed based on the different $\delta^2H_{C37}$-salinity calibrations are larger than salinity estimations which have been made based on $\delta^{18}O$ and ice volume modelling.*

**RC2:** Line 72: The authors mention that the $\delta^{18}O$ – salinity relationship may not hold through time (line 47), to use $\delta^2H_{sw}$ to reconstruct salinity, do the authors bot also have to assume this relationship holds through time? Or at least indicate here that we also do not know if this holds through time?

- **Authors:** *Yes, as described in line 47 the $\delta^{18}O$ – salinity of the open-ocean could have changed and the reviewer is right with this the $\delta^2H_{sw} - \delta^{18}O$ and $\delta^2H_{sw}$ – salinity relationship would change as well. We discuss this possibility in section 3.4, line 269-270.*

**RC2:** Line 73: Again, I would suggest the authors reconsider whether this truly speaks to regional versus local variability.

- **Authors:** *Please see our reply above.*

**RC2:** Line 81/82: This sentence is difficult to follow in the figure legend. Please consider re-phrasing.

- **Authors:** *We will rephrase it.*

**RC2:** Line 86: "… from a previously collected deeper neighboring ODP…" Be sure to give credit/indicate this was a previously collected core.

- **Authors:** *We will indicate this more clearly.*

**RC2:** Line 90: Transports cold saline waters where? Isotopically depleted water can potentially be added where?

- **Authors:** *What we mean to say is that the cold and saline water transported from the ACC into the Peru-Chile Countercurrent could be mixed with fresher and $^2$H-depleted fjord water along the southern margin.*

**RC2:** Line 91: The average salinity in the area was defined how? Based on a quarter grid perhaps?

- **Authors:** *Yes, it was based on the World Ocean Atlas quarter grid (WOA18_0.25deg_All_Years_Annual.Data). This will be indicated in the revised version.*

**RC2:** Line 97: I think you are missing a word in this sentence. This should maybe read: "… of <2 ka, and a total of 27 samples were…" Also, you might consider noting how ODP 1234 was sampled (the same way?).

- **Authors:** *We will change the sentence. We did not sample ODP 1234, the sampling is described by de Bar et al., (2018) and Weiss et al., (2019b).*

**RC2:** Line 101: I think you meant to use a comma not a period to indicate 41,000 years.

- **Authors:** *Yes, thank you.*

**RC2:** Line 103: Perhaps the word 'with' should be replaced with 'at' here.

- **Authors:** *Yes, thank you.*

**RC2:** Line 113: Please define the average differences here – is this a standard error or standard deviation etc.?

- **Authors:** *This needs to be corrected to standard deviation of replicate analysis of the same sample.*

**RC2:** Line 114: The standard deviation of what is within 0.1 per mil?

- **Authors:** *Both isotope calibration standards, NBS 19 and NFHS-1, are within 0.1‰ standard deviation of accepted values.*

**RC2:** Line 119: Indicate the model number or instrument name for the ASE.

- **Authors:** *OK. It is a Dionex 350 ASE.*

**RC2:** Line 143: RSME should be RMSE

- **Authors:** *Yes, thank you.*

**RC2:** Line 146: Are you referring to the measured salinity in the modern surface open-ocean?

- **Authors:** *The hydrogen isotope values are mostly from surface waters, however for the oxygen isotope data, we included values between 0–600 m. We will include the sampling depth information in the supplementary information.*

**RC2:** Line 149: Again, are you referring to surface ocean data?

- **Authors:** *The data points from the surface ocean dataset are from sampling depths between 0-600 m.*

**RC2:** Line 154: Is there a good citation to include here for detailing that the oxygen isotopic composition of forams depends mainly on temperature and the isotopic composition of seawater?

- **Authors:** *In the lines following line 154 we do discuss publications regarding this matter, e.g. McCrea et al. (1950) being the first laboratory study showing the relationship of oxygen isotope signal of the foraminifera with that of growth water and temperature.*

**RC2:** Line 155: "...calcite sources..." is likely not the best/most correct term. Perhaps, calcite species or polymorphs?

- **Authors:** *Yes, we will correct it to polymorphs.*

**RC2:** Line 159: "... temperature proxies and **then** obtain salinity..."

- **Authors:** *We will correct it.*

**RC2:** Line 161: "... (2011)**, where** the $\delta^{18}$O- paleotemperature..."

- **Authors:** *We will correct it.*

**RC2:** Line 162: in-situ should be italicized.

- **Authors:** *We will change it.*

**RC2:** Line 166: "...at the time of Eq. (4)..." is awkward wording, please considering clarifying.

- **Authors:** *We will clarify it.*

**RC2:** Line 167: Please define this equation, where is it from? As it stands it is not well introduced/prefaced.

- **Authors**: *Thank you. Line 166 needs to be corrected, it is referring to Eq. (5).*

**RC2:** Line 169 (Equation 6): perhaps the authors might define earlier how temperature is solved for in this equation typically? Which proxy(ies) is/are used to resolve temperature.

- **Authors:** *We explained in the Introduction how the temperature is usually resolved, see line 39-43. We made an assumption for the temperature change, since we were not able to determine bottom water temperatures, as described in the Results and Discussion at lines 195-202.*

**RC2:** Line 190: "The oxygen **isotopic signatures** of benthic Uvigernia in ODP core 1235 range between 4.15 ..."

- **Authors:** *Thank you, we will change it.*

**RC2:** Line 192: $\delta^{18}O_{foram}$

- **Authors:** *We will check the abbreviations for consistency.*

**RC2:** Line 192: Perhaps change the word "till" to "to".

- **Authors:** *We will change it.*

**RC2:** Line 193: This sentence structure needs work. "...showing an LGM to 1 ka shift of..." is difficult for a reader to follow. Again, indicate the direction of the shift.

- **Authors:** *We will change it.*

**RC2:** Line 195: This sentence structure could be improved.

- **Authors:** *We will improve it.*

**RC2:** Line 201: What is the temperature change reported in ODP 1234 – if it is consistent, please report the value(s).

- **Authors:** *We will report it clearer.*

**RC2:** Line 204: It is a little bit difficult to follow when the authors are referring to surface and benthic water isotope values – is there a clearer way to indicate benthic vs surface throughout?

- **Authors:** *We will try to make this clearer. However, we want to point out that chapter 3.1 only describes bottom seawater isotopes as stated by the title and chapter 3.2 only surface seawater isotopes. We chose this structure to make the discussion clearer and help the reader. Chapter 3.3 is comparing the seawater isotopes, oxygen and hydrogen isotope proxies, and surface versus bottom water signals – here we should make clear when we speak of bottom and when of surface water. Chapter 3.4 discusses only the global surface water signal.*

**RC2:** Line 206: Please indicate the directionality of the isotopic shift.

- **Authors:** *We will make the direction of the isotope shift clearer.*

**RC2:** Line 218: '...globally surprisingly similar in magnitude," is not very clear wording.

- **Authors:** *We will change the wording.*

**RC2:** Figure 2: General introductory statement first. "**Alkenone** hydrogen isotopic ratios measured in ODP 1235....". What does 5pt average refer to? Where are the black circles on the figure indicating $\delta^{18}O$ of the benthic foram from ODP 1234? What are the seawater temperature data reconstructed from? TEX UK37?

- **Authors:** *We noticed that it needs to be: "The green line shows the 5pt average $\delta^{18}O$ values of benthic Foraminifera from ODP 1234 (from de Bar et al., 2019)". We will edit the figure description accordingly.*

**RC2:** Line 246: Where was de Bar et al.'s work? Indicate this to the reader.

- **Authors:** *At core site 1234, Chilean Margin, this will be indicated in the updated manuscript.*

**RC2:** Line 262: I am not sure if it is commonplace to capitalize 'Recent'?

- **Authors:** *Yes, it is a noun in Geology; singular proper noun: Recent*

**RC2:** Line 269: salinity – $^2$H should be salinity- $\delta^2H_{sw}$ ?

- **Authors:** *We will change it to $\delta^2H$ to avoid confusion.*

**RC2:** Line 271: "... of a dominating **evaporative regime** on the slope of the isotope..." and on which isotope are the authors referring? Both?

- **Authors:** *We are referring to both seawater isotopes.*

**RC2:** Line 273: "Surface waters may be more sensitive **to** changes in this..."

- **Authors:** *We will change this.*

**RC2:** Line 274: salinity – $^2$H should be salinity- $\delta^2H_{sw}$ ?

- **Authors:** *We will change it to $\delta^2H$ to avoid confusion.*

**RC2:** Line 274: "... for surface waters, due to, **for example**..."

- **Authors:** *We will change this.*

**RC2:** Line 275-276: This sentence is very confusing to follow, please consider elaborating on the message here.

- **Authors:** *We will write it clearer.*

**RC2:** Table 1: Please consider adding a general statement to describe what this table is detailing. I would also consider being explicit about why the Kasper data are not included in your discussion.

- **Authors:** *We will add this.*

**RC2:** Line 297-298: Please consider adjusting the "Globally distributed..." sentence as it is difficult to follow. Subscript C37 ($C_{37}$).

- **Authors:** *We will change this.*

---

## Author Response (AR1)

**Glacial-interglacial seawater isotope change near the Chilean Margin as reflected by $\delta^2H$ of C$_{37}$ alkenones.**

Katrin Hättig[1], Devika Varma[1], Stefan Schouten[1,2], Marcel T. J. van der Meer[1]

[1] Department of Marine Microbiology and Biogeochemistry, NIOZ Royal Netherlands Institute for Sea Research, Texel, the Netherlands.

[2] Department of Earth Sciences, Faculty of Geosciences, Utrecht University, Utrecht, the Netherlands.

*Correspondence to*: Katrin Hättig (katrin.haettig@nioz.nl)

**Authors response:** *We thank the editor and reviewers for their constructive comments on the manuscript, which helped to improve it. Please find a detailed reply to all comments below.*

**Editor**

Dear authors,

Thank you for your thorough responses to the Reviewer's comments. You have made responses which indicate that the concerns of the reviewers can be addressed in a revised version of the manuscript. Although many of the comments are minor (typos, clarifications) there are some suggested changes which may take a bit more work, including showing the uncertainties in more detail and carefully editing the Discussion to ensure your arguments are clear. Although there are no major issues with your data nor interpretations, care will be needed to ensure that all of the reviewer concerns can be addressed in a revised version. As a result, I have recommended major revisions to acknowledge the work required.

Some minor comments:

Line 21 Reviewer 1 - the use of physiochemical or physicochemical. I agree with your use of the term, and suggest that you consider whether any of these factors in your response to the review might be usefully added to the main text, so ensure that the message is clear to a reader?

Lines 107-108 Reviewer 1: please also check the journal guidance about whether kyr or ka is appropriate for durations of time.

Line 9 Reviewer 2: regional vs local. I agree that line 9 does not need to change, and I also follow your argument about the sites being different and so >1 core site might extend beyond the local. Yet, Figure 1 illustrates the issue: the two sites are "the same" on this map, which shows the "region"? I suggest that you carefully assess each of your instances of "regional" interpretations, and consider whether alternative terms might be more precise for each of those specific points (e.g. complementary sites, or contrasting deep water variations).

- **Authors:** *We thank the editor for the positive recommendation and have addressed the reviewers concerns in the revised version and our rebuttal. We revised our calculations on the uncertainties and show them now in a hopefully clearer way to the reader. We checked the Journals guidelines and applied them in the revised version (e.g. Ka versus Kyr) and we carefully checked the use of "regional" versus local in our manuscript.*

**Reviewer 1:**

**General Comments**

**RC1:** In this paper, Hättig et al. generate a paired surface-benthic ocean temperature / hydrogen / oxygen isotope record from the Chilean Margin spanning the last 40 kyr. Their new data are similar to those from a neighboring site and they also show that the mismatch in planktic-benthic seawater isotope change is consistent to that observed in the handful of other extant LGM-modern alkenone hydrogen isotope records. This multi-proxy approach is creative and demonstrates the potential of the alkenone $\delta 2H$ proxy. This manuscript also shows a good awareness of prior work and current technical and analytical limitations. I believe it is well within the purview of Climate of the Past and therefore recommend it for publication following extensive but relatively minor revisions.

My first suggestion is a careful grammatical overhaul; I am sure I did not identify every single instance where the language and logical flow could be improved and I would encourage the authors to actively seek out opportunities to do to so rather than merely addressing those listed below.

- **Authors:** *We thank the reviewer for the technical corrections pointed out. We took the opportunity to go through the text carefully. We tried to improve the consistency and flow by e.g. "seawater" one word and correct use of dashes and hyphens following the journal guidelines.*

My other main recommendation would be a more extensive treatment of the uncertainties. This work involves several transfer functions (i.e., $\delta 18O$ to $\delta 2H$, $\delta 18O$ to salinity, foraminiferal $\delta 18O$ to seawater $\delta 18O$, alkenone $\delta 2H$ to seawater $\delta 2H$), calibration error, and general unknowns (benthic paleotemperatures); the authors do a good job of bringing these to our attention, but I believe a slightly more quantitative assessment of how these compounded uncertainties (roughly calculated) affect their final conclusions would be beneficial. (E.g., does this mean that the last deglaciation is the smallest change we can reasonably do this type of analysis for? Is the surface-depth isotope offset within error?)

- **Authors:** *We addressed this in the section 3.4 where we discuss the hydrogen isotopes versus the oxygen isotopes in a global perspective. Regarding the uncertainty*

*in benthic paleotemperatures, we performed a sensitivity analysis and refer to our rebuttal below for the specific comments on this topic. We now describe in the main text the propagation of errors when the different transfer functions are used where we mainly focused on the error in the calculation of differences between LGM and Recent, rather than absolute values. For example, the error of absolute $\delta^2 H_{SSW}$ estimations is the propagated error of the $\delta^2 H_{SSW}$ calibration (e.g. 5.8‰ for the Gould calibration) and analytical error in absolute $\delta^2 H_{C37}$ values (5‰ based on replicate analysis of Mix B standards), i.e. 7.6‰. However, the calculation of the difference in $\delta^2 H_{SSW}$ between LGM and Recent is the propagated error in replicate analysis of two $\delta^2 H_{C37}$ analysis (e.g. 3 and 1, i.e. 3.1‰) and the error in the slope of the $\delta^2 H_{SSW}$ calibration (e.g. 0.4 for Gould et al., 2019). Thus, for this particular example we added in section 3.2 the following sentence: "Conversion of the shift in $\delta^2 H_{C37}$ of -20 ±3‰ into $\Delta\delta^2 H_{SSW}$, using the calibration of Gould et al. (2019) and considering the error in the slope, results in a shift of -14 ±4‰ from LGM to the most recent sample. Application of the core-top calibration from Mitsunaga et al. (2022) (Eq. 8) gives an identical change of -14 ±3‰."*

**Specific Comments**

**RC1:** Lines 52-56: My understanding was that the alkenone-seawater hydrogen isotope fractionation's salinity-dependence was observable in culture (Schouten 2006 and your other references) but not always in sediment (Weiss et al., 2019; Mitsunaga et al., 2022). I think you can say this and acknowledge that effect of salinity on alkenone-water fractionation is not settled.

- **Authors:** *Yes, the impact of salinity is unclear, we rephrased these lines and state that the different responses or absence of response of $\delta^2 H$ to salinity make quantitatively constraining past salinity changes difficult.*

Site 1235; explicitly call out the benefits of having the nearby Site 1234 δ2HC37 and temperature records, then get into why your site and approach are different (which you do: different depths, etc.). Also, in general, especially in your abstract and introduction, you can do a better job saying why your study is so important and why you needed to make these measurements so close to an existing δ2HC37 A sentence or two here or there could make a big difference.

- **Authors:** *We added text at the different parts of the revised manuscript making it clearer why we chose site 1235 and how it is beneficial.*

**RC1:** Lines 94-97: Were planktic foraminifera unavailable? That could be another means of reconstructing surface water δ18O. And then you could avoid the δ18O- δ2H transfer function, which is an additional source of error…

- **Authors:** *Planktic foraminifera were picked when abundant in core slides and their $\delta^{18}O$ was analysed. However, this record did not cover the whole 20 ka period sufficiently enough to calculate the surface oxygen isotope water shift during the last deglaciation.*

**RC1:** Lines 128-130: How different were the δ2HC37:3 and δ2HC37:2 values? I.e., was there any C37:3-C37:2 (inter-alkenone) fractionation? I think you should mention it in your results; I know it is not the main thrust of this paper, but some people (Sachs, D'Andrea) have observed an offset while others (Weiss, Mitsunaga) have not, and this could be of interest to the biochemists studying coccolithophore cell water dynamics. At the very least, if there is no (statistically significant) offset, mentioning that legitimizes your choice to report combined δ2HC37.

- **Authors:** *Yes, to account for inter-alkenone fractionation we integrated $\delta^2H_{C37:3}$, $\delta^2H_{C37:2}$ separately as well as combined peaks of $C_{37:2}+C_{37:3}$ alkenones, $\delta^2H_{C37}$. Our analyses allowed us to do so in one measurement. All values are reported in the data file on Pangaea (https://doi.org/10.1594/PANGAEA.958880) and we added to the supplement file the individual $C_{37}$ alkenone $\delta^2H$ values. The $C_{37:3}$ alkenone suggests a $\delta^2H$ shift of 24‰ and the $C_{37:2}$ alkenone a shift of 16‰. However, van der Meer et al. (2013) have shown that a weighted average or combined integration of $C_{37}$ peaks might be more appropriate than individual values when using alkenone hydrogen isotopes for reconstructing paleo sea surface salinity changes. An additional reason for using the combined $C_{37}$ alkenone values is that older datasets are all based on the integrated $C_{37}$ alkenones and we compare our results to these datasets (e.g. Pahnke et al., 2007; Kasper et al., 2014). Lastly, the published $\delta^2H_{C37}$ - $\delta^2H_{SW}$ calibrations (Mitsunaga et al., 2022; Gould et al., 2019) are also based on the combined $C_{37}$ peak, and we therefore prefer to stick to this. We made changes in the Introduction and Methods to clarify that we focus on long-chain alkenones with 37 carbons with two and three double bonds and their combined hydrogen isotope signal.*

**RC1:** Lines 137-150: I think an obvious question is why bother with converting δ2HC37 to δ2HSW to salinity if there are existing δ2HC37-salinity calibrations, and I think you need to address this, at least briefly. What could be said is that the direct salinity calibrations are worse than the δ2HC37 to δ2HSW ones in core-tops (Mitsunaga, Weiss). Of course, then you have to explain why δ2HC37-salinity is worse than δ2HC37-δ2HSW if the δ2HSW-salinity relationship is so tight...

- **Authors:** *As we mentioned in the Introduction at line 55-69, culture $\delta^2H_{C37}$-salinity calibrations are based on single species and fixed conditions, which could be problematic when translating to environmental calibrations. The core-top calibration should be the most suitable since it includes multiple species and environmental variability including sedimentation and to some degree preservation. However, reconstructed past salinity changes become very large when applying these $\delta^2H$-*

*salinity relationships, suggesting that we are missing a piece of the puzzle (discussed in Weiss et al., 2019b). The SPM calibration of $\delta^2H_{C37}$ to $\delta^2H_{SW}$ also includes contributions from different species and the effects of other environmental parameters and fits very well with the updated core-top calibration of Mitsunaga et al. (2022). With the reconstructed $\delta^2H_{SW}$ information, salinity interpretations can carefully be made with a global (as demonstrated) and local waterline. In the revised version we changed the Introduction lines 70-75 stating clearer why we choose this approach and the problem with the direct salinity translation.*

**RC1:** Lines 195-208: I get that you were unable to measure benthic paleotemperatures, but that does introduce some holes in your argument. It is good that you have the Schrag et al. bottom-water temperatures, because otherwise I would be skeptical that planktic and benthic warming would be the same. Is there any way you could show how much the uncertainty in your calculations matters? I.e., if you are off by X °C, how much does it matter to your calculations of benthic δ18OSW change? And then how does that error propagate to δ2HSW calculations?

- **Authors:** *We performed a sensitivity analysis (see below) which showed that one degree of temperature change affects the $\delta^{18}O_{SW}$ shift by 0.2‰ and the $\delta^2H_{SW}$ by 1.4‰. In fact, the sensitivity analysis shows that even with negligible bottom water temperature change, the magnitude of $\delta^2H_{SW}$ change of bottom waters will still be lower than that of surface waters ($\delta^2H_{SW}$ = 14‰).*

  *The following table shows the impact of different bottom water temperature changes (ΔT) on the bottom water oxygen and hydrogen isotope change at ODP Site 1234 and 1235. $\Delta\delta^{18}O_{foram}$ estimate is based on the average observed at sites 1234 and 1235 (see Table 1 in preprint). For the propagated error calculation (based on the slope error), we refer to the answer we gave above.*

| $\Delta\delta^{18}O_{foram}$ | ΔT | $\Delta\delta^{18}O_{SW}$ | $\Delta\delta^2H_{SW}$ |
|---|---|---|---|
| 1.6 ±0.2‰ | 0 °C | 1.6 ±0.2‰ | 10.6 ±2‰ |
| 1.6 ±0.2‰ | 1 °C | 1.4 ±0.2‰ | 9.3 ±2‰ |
| 1.6 ±0.2‰ | 2 °C | 1.2 ±0.2‰ | 7.9 ±2‰ |
| 1.6 ±0.2‰ | 3 °C | 1 ±0.2‰ | 6.5 ±2‰ |
| 1.6 ±0.2‰ | 4 °C | 0.8 ±0.2‰ | 5.1 ±2‰ |

*In the revised version, line 278-283, we added a sentence addressing this:" Potentially, we underestimated the change in the latter due to a lack of reconstructed bottom water temperatures. However, sensitivity analysis shows that even with much smaller bottom water temperature changes than used here, the magnitude of $\delta^2H_{BSW}$ change will still be lower than that of $\delta^2H_{SSW}$. For example, in the unlikely scenario that bottom water temperatures have remained constant, the maximum change of $\Delta\delta^{18}O_{BSW}$ is still only -1.6 ±0.2‰, or -10.6 ±2‰ of $\Delta\delta^2H_{BSW}$. Thus, it seems likely that at*

*the Chilean Margin the glacial–interglacial change in surface water isotopes was larger than that of bottom water isotopes."*

**RC1:** Lines 249-259: Makes sense – if you assume the same LGM-modern temperature change, you get similar changes in benthic δ18O as global. If you talk about temperature uncertainties earlier, this might be a good time to discuss what you discovered. What if deglacial benthic temperature change (and therefore δ2HSW change) was more / less than expected based on Schrag et al. numbers? Does it come close to affecting your argument that planktic δ2HSW change was much greater than benthic?

- **Authors:** *If the bottom water temperature change was higher than the assumed 4 ⁰C, this would result in a smaller seawater isotope change and therefore smaller bottom water salinity change. Even if the bottom water temperature change is close to 0 ⁰C, then the seawater isotope change of the bottom waters is still smaller than that of surface waters. We addressed this issue in line 278-283, see also comment above.*

**RC1:** Line 261-281: I think you need to qualify this a bit. It is a very interesting trend if true, but since there are only seven records—including the anomalous Mozambique channel Kasper et al. one—from mostly low to mid-latitudes, as you acknowledge, I am not sure the data is there to do so quite yet. Presumably, there are other Mozambique channel-type sites out there, we just have not discovered them yet. I think you need a sentence or two at the start of this section acknowledging the small number of extant records. But as long as the trend is real…

- **Authors:** *We agree that the published records do not cover the entire globe and definitely not equally, though we note that it will be some effort to generate substantially more records. We changed line 294-299: "To examine the global distribution of hydrogen isotopes during the last deglaciation we compare the two Chilean Margin records to published global alkenone hydrogen isotope records (Table 1). These comprise five low to mid–latitude records of which four show a negative glacial–interglacial shift while the record from the Mozambique Channel of Kasper et al., (2015) reflects a positive change. The latter may be due to the close location and changing distance to a river mouth, with high BIT index, and therefore this record is not further considered here."*

**RC1:** Lines 268-269: I think it is believable that the surface ocean would have freshened first, but I think it is intriguing that these changes would not have propagated to the deep ocean within several thousand years given the mean mixing time of seawater. Do you have any other sources that could support this?

- **Authors:** *We are not aware of any other studies which could support this. We agree to the reviewer that due to mixing over thousands of years we would expect a propagation of the surface changes eventually towards the bottom.*

**RC1:** Lines 274-275. I think you could come right out and say that the LGM was drier than the modern. A source or three here could really strengthen your argument and / or I think you need to flesh out the mechanisms by which a drier atmosphere could change the salinity- δ2H relationship, which then more directly affects the surface than the depths. (That last part makes perfect sense to me.)

- **Authors:** *We thank the reviewer for this suggestion. We did not find a good source and it seems to be a controversial topic in the literature, as some parts of the Earth may have been wetter and some drier during the LGM. We therefore wrote our discussion carefully.*

**RC1:** Table 1: I think you need to explain in the text or table caption somewhere why the Mozambique channel Kasper record is excluded from your calculations (i.e., changing riverine inputs rather than seawater δ2H change). I would also use "source" instead of "paper" for the leftmost column.

- **Authors:** *This comment prompted us to revise the format of Table 1 substantially so that all the data and sources are clearer. In the table we changed "paper" to "source" and rewrote the table caption. In the beginning of section 3.4 we added text stating why we did exclude the Mozambique channel record (Kasper et al., 2015) for the further interpretation of the LGM-Recent hydrogen isotope shift.*

**Technical Corrections**

**RC1:** Line 2: [Here, and subsequently] I do not have strong feelings about this, but I have been told that delta notation—i.e., "δ2H" or "δ18O"—is an adjective and therefore that it always needs to be followed by a noun, i.e., "δ2HC37 values," "this δ2HC37 record," "our new δ2HC37 data," etc.

- **Authors:** *We checked this.*

**RC1:** Line 2: [Here, and subsequently] I believe that the number of carbons should be subscripted, i.e., "C37."

- **Authors:** *We changed this.*

**RC1:** Line 12: "Shift" is overused throughout the abstract, try to find synonyms.

- **Authors:** *We improved this by using the synonym "change".*

**RC1:** Line 12: The last / most recent deglaciation?

- **Authors:** *We added "the last".*

RC1: Line 16: Separate "waterline" into two words.

- **Authors:** *We changed this.*

**RC1:** Line 21: "Physiochemical"

- **Authors:** *We do mean to say physicochemical, as it refers to both physical and chemical attributes e.g., temperature, salinity, pressure, density.*

**RC1:** Line 26: "Local climate regime" is vague; do you mean temperature?

- **Authors:** *We mean different climate parameters such as precipitation, temperature and humidity.*

**RC1:** Line 28: "Low oxygen and hydrogen isotopic values" is vague, could be referring to D/H ratios, etc. If you mean "δ2H or δ18O values," just say so. (You do this in lines 30-31 anyway.)

- **Authors:** *Yes, we mean $\delta^{18}O$ and $\delta^2H$ values and made changes to clarify this.*

**RC1:** Line 36: The "f" in "δ18Oforam" is sometimes capitalized, sometimes not. I would go with lowercase, but just be consistent. You also re-abbreviate this (Line 95) elsewhere; you only need to once.

- **Authors:** *We made it consistent and use $\delta^{18}O_{foram}$.*

**RC1:** Line 36: [Here, and elsewhere?] You already introduced an abbreviation for the oxygen isotopic composition of seawater (δ18Osw), might as well use it.

- **Authors:** *We changed this.*

**RC1:** Lines 43-45: Also LGM porewater δ18Osw measurements; see Adkins & Schrag 2001.

- **Authors:** *We added porewater and the reference.*

**RC1:** Line 45: "To derive" should be "deriving."

- **Authors:** *We changed this.*

**RC1:** Line 61: What is the expected deglacial salinity change?

- **Authors:** *We added in the introduction that based on this data, oxygen isotopes, and modelling attempts the expected salinity change is 1-2 psu.*

**RC1:** Line 70: "Alkenone" should be "alkenones."

- **Authors:** *We changed this.*

**RC1:** Figure 1: The location of the word "Chile" is misplaced. I get not wanting to put it near where Chile actually is, because it's a bit crowded there, but maybe that means it is unnecessary. There are also several typos in the caption.

- **Authors:** *We removed the word "Chile".*

**RC1:** Line 85: [Here, and elsewhere] I would mostly use "site" instead of "core" in this section since you are referring to the physical locations of the cores rather than the cores themselves. I would use "core" if you were discussing the characteristics of the core (i.e., length, lithology, etc.).

- **Authors:** *We thank the reviewer for this suggestion and we change it accordingly.*

**RC1:** Line 86: Remove "located."

- **Authors:** *We removed it.*

**RC1:** Line 88: Add a comma between "upwelling" and "stimulating."

- **Authors:** *We added a comma.*

**RC1:** Line 96: [Here, and elsewhere] be consistent with using a comma after "i.e." or "e.g." I believe the difference is British versus American English; see which the journal follows.

- **Authors:** *Indeed, the difference is American English versus British English. We follow in the text British English and we therefore removed the commas.*

**RC1:** Lines 96-97: This is a run-on sentence.

- **Authors:** *We changed the sentence.*

**RC1:** Line 106: Misplaced parenthesis.

- **Authors:** *We changed this.*

**RC1:** Lines 107-108: "Ka" should be "kyr." (Ka [or Ma] specifically means years ago, where here I think you just mean x cm per x thousand years.) Also, you do not need the periods between "cm" and "kyr."

- **Authors:** *We changed this accordingly.*

**RC1:** Line 111: Check if you need the hyphens here. I am not sure that you do.

- **Authors:** *We deleted the hyphens.*

**RC1:** Line 114: Is the standard deviation referring to NBS-19 or the internal standard or both?

- **Authors:** *We added that it is referring to both. We always try to have both standards within 0.1‰. NBS19 is the official IAEA reference standard and is very homogeneous. The NFHS1 in house standard is also foraminifera material, but less homogeneous.*

**RC1:** Line 115: Both might be acceptable, but I have more commonly seen "per mille." Also, I do not think you need to define the ‰ symbol, but if you do, you should do so the first time you use it, not here.

- **Authors:** *We changed this.*

**RC1:** Line 133: [Here, and elsewhere] to clarify, "online" refers to the value measured in the lab that day and "offline" refers to the published / "correct" values (via Schimmelmann or others)?

- **Authors***: The reviewer is right. We changed it to: "The performance and stability of the instrument were monitored by measuring a standard containing 15* n-*alkanes at different concentrations (Mix B, A. Schimmelmann, Indiana University), at the start of each day. Samples were only run when the average difference between the measured values for the Mix B standard and the certified values for this standard and their standard deviation were less than 5‰."*

**RC1:** Lines 134-135: [Here, and elsewhere] be consistent with spacing between mathematical symbols (±, <, ‰, etc.). Check journal-specific conventions.

- **Authors:** *We checked this throughout the manuscript and followed: between symbol and number no space e.g. -173, ±5, between number and ‰ no space as it is not a unit. Between number and unit space.*

**RC1:** Line 144: "RSME" should be "RMSE."

- **Authors:** *We changed this.*

**RC1:** Line 145: "Waterline" should be two words.

- **Authors:** *We changed this.*

**RC1:** Line 162: Italicize "in situ," but I cannot tell what it adds here. (Do you mean from sediment versus from the water column or culture?) Try "estimated using the benthic genera Cibicidoidies and Planulina."

- **Authors:** *The equation by Lynch-Stieglitz et al. (1999) as rearranged by Cramer et al. (2011) is calculated from core-top samples, i.e. a field calibration. Pearson et al. (2012) described this calibration as* "in situ". *We describe it in the revised version as field calibration.*

**RC1:** Line 163: Not sure these are recent. "Subsequent," maybe? Also, phrased a little awkwardly ("other species' values are projected onto the Cibicidoidies and Planulina function?").

- **Authors:** *We changed it.*

**RC1:** Lines 251-252: I am wondering if you could use separate abbreviations for surface and deep water δ2HSW values and change – δ2HSWp and δ2HSWb? I am sure you can think of something better and / or do not be afraid to use synonyms for "surface" and "deep" ("planktic" and "benthic," etc.) too.

- **Authors:** *We think it is good to stick to surface and bottom water, this is how the article from Results to Discussion is structured and we think this will be clear to the proxy and modelling community. We hesitate to use planktic and benthic as it could lead to more confusion, and we like the reader to focus on the water masses, rather than the proxies. In the revised manuscript we now use the abbreviation $\delta^2 H_{SSW}$ for*

*surface seawater and $\delta^2H_{BSW}$ for bottom seawater. This is similar to the abbreviations used for sea surface and bottom temperature i.e. SST and BWT.*

**RC1:** Line 262: "Remarkable" should be "remarkably," although I would probably just take it out.

- **Authors:** *We changed this to remarkably.*

**RC1:** Line 265: "Waters" typo.

- **Authors:** *We changed this.*

**RC1:** Line 269: "Salinity-δ2H" instead of "salinity-2H?"

- **Authors:** *We changed it to $\delta^2H$ to avoid confusion.*

**RC1:** Line 285: "Std" = standard deviation?

- **Authors:** *Yes, with Std we mean the standard deviation and expanded the abbreviation.*

**Reviewer 2:**

**General Comments**

RC2: Hättig et al. present an interesting new collection of paired surface ocean $\delta^2H_{C37}$ and benthic $\delta^{18}O_{FORAM}$ records from the Chilean Margin for reconstructing changes in ocean isotopic composition ultimately speaking to water mass and/or salinity variability from the LGM.

The data show good agreement with a previously published record from the same locality, and the treatment of sample and data analyses are consistent with those accepted in the literature. The authors' new approach/application of hydrogen isotopic composition of alkenones for reconstructing the hydrogen isotopic composition of surface water (and then ultimately of salinity) is unique and interesting and certainly warrants publication. I believe this body of work is suited to Climate of the Past, following careful revision.

My overall recommendation to the authors would be to carefully consider reviewing the paper for clarity and detail. There are many instances of grammatical errors/sentences that are difficult to follow. In particular, the discussion and even more so the conclusion is not as well organized as it could be. The conclusion should be a thoughtful overview of the major findings of the work and as it is written it is relatively difficult to follow. I strongly encourage the authors to review the organization of this section and to also improve on the overall portrayal of the major conclusions from this work.

- **Authors:** *We thank the reviewer for the positive response. We carefully went through the text.*

Next, if the authors could be clearer and more explicit about the errors that they are reporting throughout, it would be a more powerful reflection of the data. There are of course assumptions that must be made throughout; however, I also wonder if the authors would consider reporting the error in the transfer functions used (i.e., Equations 1 through 8).

- **Authors:** *We agree with the reviewer that clear reporting of data and the propagation of errors is important. We, therefore, display the dataset of the modern open-ocean seawater isotopes and salinity in figure S2 and S3 of the supplemental information. The root-mean-square errors (RMSE) for the transfer functions (1, 7, 8) give an indication of the errors for the reconstructed $\delta^2H_{sw}$ from $\delta^2H_{C37}$ or $\delta^{18}O_{sw}$. However, since we are mainly concerned with calculating differences (between LGM and Recent) then absolute values, we use the errors associated with analysis of differences in $\delta^2H_{C37}$ or $\delta^{18}O_{foram}$ as well as the errors in the slope of the difference transfer functions. When appropriate we tried to make the error propagation clearer to the reader in the different sections.*

RC2: Because much of the discussion and conclusions deal with the offset found between surface and benthic estimations/reconstructions of $\delta^2H_{sw}$, and because the reconstruction specifically of benthic $\delta^2H_{sw}$ relies on temperature, which the authors do not model, I wonder if the authors considered running (and then reporting on) an experiment where they vary the bottom temperature estimates for the benthic reconstruction of $\delta^2H_{sw}$ to determine a conceivable (appropriate) range (and to demonstrate the sensitivity) of this particular offset (Lines 195-208). It is difficult to follow that the authors assume temperature is consistent at depth when they report a major shift in isotopic/salinity signatures between surface and benthic environments. Whether this can be resolved here or not, I do believe the authors should spend some time in their discussion on this discrepancy. Are there no planktonic foraminifera $\delta^{18}O$ data to use for reconstructing surface salinity?

- **Authors:** *We thank the reviewer for this comment which is identical to a comment made by reviewer 1. We, therefore, refer to our reply to reviewer's 1 comment where we perform a sensitivity analysis which shows that even with unrealistically small bottom water temperature changes, the change in $\delta^2H_{sw}$ of bottom waters would still be lower than that of surface waters. We added a statement in the discussion in section 3.4.*

RC2: Finally, I wonder why the authors did not test the $\delta^2H_{C37}$ – salinity calibration models for reconstructing salinity at this site (at the very least plotting their new $\delta^2H_{C37}$ data alongside previously published data). The authors mention this calibration in the literature; however, the paper overall might be made more powerful and rigorous by including this evaluation.

- **Authors:** *The $\delta^2H_{C37}$ –salinity calibrations based on culture studies (e.g., Schouten et al., 2006.), core-top studies (Weiss et al., 2019) and SPOM filters (Gould et al., 2019) have been compared and discussed for the ODP Core Site 1234 at the Chilean Margin in the study of Weiss et al. (2019b). Our hydrogen isotope data from site 1235 is very similar to the last deglaciation $\delta^2H_{C37}$ shift reported by Weiss et al. (2019b) and other global sites. Therefore, we did not see the need to repeat this discussion here, though we briefly summarize it in the revised manuscript.*

**Specific Comments:**

RC2: Line 9/the overarching argument that the work presented illustrates a 'regional' not 'local' phenomena/record: I suppose it is relatively subjective, however, it is my experience that a regional phenomenon (especially in the vast ocean environment) would require the authors to have evaluated core samples taken from a much larger range of latitude/locations along the Chile margin. In my experience, 12km distance between two cores is showing consistency at one specific local, where the two cores would be treated essentially as replicate records to illustrate a robust finding. I suggest the authors reconsider whether their work presented is truly extending a local record to a regional one with just two cores so close in proximity. Although this does not detract from the importance of their record, nor does it negate the impact two core records has on improving the results.

- **Authors:** *The reviewer is correct that the distance between the two cores is only 12 kilometers, but the water depth from which the cores are retrieved is quite different and therefore still represents somewhat different conditions. Thus, although the cores are from very close proximity, we feel that "local" does not entirely do this study justice, but we agree with the reviewer that "regionally" is perhaps making it too big. However, we do note that a large number of palaeoceanographic studies draw regional conclusions by extrapolating results from a single core to a whole region. Here we extrapolate the results of two replicate records to a whole region which is, in our view, more robust than most other studies. Furthermore, the comparison to other records confirms that the signal is not reflecting local circumstances (e.g. river input) but regional and perhaps global changes. Therefore, as also indicated by the editor, we prefer to keep using regional in the title, but carefully checked the manuscript where this is appropriate to use.*

RC2: Throughout the manuscript, I would also suggest the authors be very clear and careful about describing isotopic trends/directionality. Whenever a change is indicated in isotopic composition through time, it would be very helpful to the reader to know in which direction the change occurred (i.e., the seawater has becoming enriched in $\delta^{18}O$ for example). This is an important detail that will aid the reader throughout.

- **Authors:** *We made this clearer by adding the direction of the shift more often e.g. negative glacial-interglacial shift. Furthermore, we updated Table 1 considerably, including making the direction of the shifts clear.*

RC2: Figure 2 and Table 1: The figure and table headings here are a little bit difficult to follow. I suggest (as was done nicely in Figure 1) the authors start the headings with a more general/overarching statement of what the figure/table are describing/illustrating before describing specific aspects of the figure/table. Ensure that the figure/table titles can be read independently of the main text and be fully understood.

- **Authors:** *We changed the captions of Figure 2 and Table 1.*

**Technical Corrections:**

RC2: Line 2: subscript on C37 ($C_{37}$)

- **Authors:** *We have corrected it.*

RC2: Line 9 and throughout the work: please determine whether you will spell the word 'seawater' as one word (as in my experience it is often spelled) or as two words 'sea water'. There are many instances of both iterations throughout the manuscript, and this should be cleaned up before publication.

- **Authors:** *We made it consistent.*

RC2: Line 12: It is important I think to indicate that the second core reported on in this work is indeed a previously reported on/published record - this is good to indicate even in the abstract here and then also throughout.

- **Authors:** *We made this clearer in the revised version. Both in the Abstract (added clearer in the revised version) and in the Introduction we state that we present a new hydrogen isotope record of ODP core 1235. In the introduction line 76-77, as well as throughout the methods and results (e.g. line 210 and 213), we state clearly that we compare our record to published data of ODP site 1234 (i.e. de Bar et al., 2018 and Weiss et al., 2019b).*

RC2: Line 16: Make it clear here that "this suggests a shift of ca. 5 per mil in benthic/deep water".

- **Authors:** *In the revised manuscript we made this clearer and introduced the abbreviation $\Delta\delta^2H_{BSW}$ for bottom seawater. As stated in the reply to the comments of reviewer 1, we prefer to use the term bottom water, rather than deep or benthic.*

RC2: Line 21: again be clear about how you spell seawater – perhaps here the parentheses are not necessary and just spell out 'seawater'.

- **Authors:** *We revised the manuscript for consistent use of "seawater".*

RC2: Line 22: the phrasing 'which could improve' is a bit weak, perhaps just 'which improve our understanding'...

- **Authors:** *We changed it.*

RC2: Line 28: Be sure the use of the colon ":" throughout is necessary. For instance, this sentence could instead read: "... isotopes of seawater, where during glacial periods large amounts of water...."

- **Authors:** *We corrected it.*

RC2: Line 28: "Low oxygen and hydrogen isotope values" is not clear, do you mean more negative? It may be helpful to include/define the equations for $\delta^{18}O$ and $\delta^2 H$ here. Since $\delta^2 H$ especially is a relatively 'new' proxy for seawater salinity, defining the hydrogen isotopes explicitly could be informative.

- **Authors:** *We added "more negative". We discuss the updated equations for $\delta^{18}O$ and $\delta H$ in the Material and Methods section.*

RC2: Line 29: You might consider removing the word "the" in this sentence: "....ice on land, leaving seawater more enriched in heavier deuterium isotopes."

- **Authors:** *We corrected it.*

RC2: Line 30: Awkward wording choice to describe 'leading the formation of ...'. Perhaps something along the lines of: "...the decrease in global ice volume releases low density fresh water with relatively low..." Also, consider throughout the manuscript when discussing isotopic values whether you are being consistent in your description of 'low vs high' or 'heavier vs light' or 'enriched' etc. isotopic signatures. Choosing one way to describe the directionality of the isotopes is much clearer.

- **Authors:** *Agree, we use low and high values and highlighted in the revised manuscript that low means more negative/ less positive.*

RC2: Line 31: seawater

- **Authors:** *We changed it.*

RC2: Line 32: "... have a close relationship..." is perhaps not the most technical way to describe this, perhaps something more like "... are correlated with..." or "...are tightly coupled with..."

- **Authors:** *We changed it to tightly coupled with...*

RC2: Line 33/35: seawater (please check this throughout, I won't call it out in my further comments)

- **Authors:** *Thank you for pointing this out, we checked its consistent use.*

RC2: Line 36: I was always taught to refer to the "isotopic" composition of something... so perhaps "... is a function of the oxygen isotopic composition of ambient seawater..."

- Authors: *We checked the text.*

RC2: Line 41: "… or clumped carbon isotopes ($\Delta_{47}$)…" for those unfamiliar with $\Delta_{47}$.

- **Authors:** *Thank you, we changed it.*

RC2: Line 41: "In some cases, independent temperature proxies based on organic…" The authors might consider including the word 'independent' here to indicate they are proxies not coming from forams themselves but other sources.

- **Authors:** *We added the word independent.*

RC2: Line 43: "…records and modelling…" is a bit awkward wording. Please consider re-evaluating this sentence structure.

- **Authors:** *We changed it.*

RC2: Line 46/47: "… relationship between $\delta^{18}O_{sw}$ and salinity." The 'salt content' is strange wording.

- **Authors:** *We changed it to salinity.*

RC2: Line 47: Perhaps the last sentence would be clearer if written something like: "However, it is uncertain whether this relationship holds through time."

- **Authors***: We followed the reviewers suggestion.*

RC2: Line 50: Again, please be clear on the direction of the isotopic change.

- **Authors:** *We indicated the negative direction of the shift.*

RC2: Line 51: Perhaps define $C_{37:2}$ and $C_{37:3}$ here because you do not define this 'shorthand' earlier. Since you are opting to combine the signals, it is important to describe to the reader what these two parameters are.

- **Authors:** *We indicated that the hydrogen isotope composition of haptophyte algae is derived from long chain alkenones with 37 carbon atoms with two and three double bonds combined ($\delta^2 H_{C37}$).*

RC2: Line 57: What $\delta^2H$ signature are you referring to here? The algal $\delta^2H$ response to changes in salinity? I believe here is where the authors describe the range in salinity reconstructions possible when using culture vs SPOM vs core top $\delta^2HC37$ – salinity calibration models. I do think it would be a powerful discussion to include a small discussion on what the data in this new core would show if a $\delta^2H_{C37}$-salinity calibration model was used to reconstruct salinity directly from $\delta^2H_{C37}$. Especially given the authors argue that the dominant species of alkenone producers in this region are likely to be *E. hux*

- **Authors:** *We refer to the alkenone $\delta^2H$ and state this now clear in the revised manuscript. The reviewer's idea was already extensively discussed by Weiss et al. (2019b). We therefore refer the reader to that study and only briefly summarized it in*

*the current manuscript (see response to the earlier comment on this topic). We slightly extended this summary in the revised version, line 59-77.*

RC2: Line 60/61: This sentence is difficult to follow, larger than expected based on what?

- **Authors**: *We rephrased this and added another sentence, see line 70-77. We mean to say that the salinity changes Weiss et al. (2019b) reconstructed based on the different $\delta^2H_{C37}$-salinity calibrations are larger than salinity estimations which have been made based on $\delta^{18}O$ and ice volume modelling.*

RC2: Line 72: The authors mention that the $\delta^{18}O$ – salinity relationship may not hold through time (line 47), to use $\delta^2H_{sw}$ to reconstruct salinity, do the authors bot also have to assume this relationship holds through time? Or at least indicate here that we also do not know if this holds through time?

- **Authors:** *Yes, as described in line 47 the $\delta^{18}O$ – salinity of the open-ocean could have changed and the reviewer is right with this the $\delta^2H_{sw}$ – $\delta^{18}O$ and $\delta^2H_{sw}$ – salinity relationship would change as well. We discuss this possibility in section 3.4, line 269-270.*

RC2: Line 73: Again, I would suggest the authors reconsider whether this truly speaks to regional versus local variability.

- **Authors:** *Please see our reply above.*

RC2: Line 81/82: This sentence is difficult to follow in the figure legend. Please consider re-phrasing.

- **Authors:** *We will rephrase it.*

RC2: Line 86: "… from a previously collected deeper neighboring ODP…" Be sure to give credit/indicate this was a previously collected core.

- **Authors:** *Both cores ODP 1235 and ODP 1234 were collected during the same ODP expedition Leg 202 which is cited with Mix et al., 2003. We added the Leg 202 number.*

RC2: Line 90: Transports cold saline waters where? Isotopically depleted water can potentially be added where?

- **Authors:** *We changed this.*

RC2: Line 91: The average salinity in the area was defined how? Based on a quarter grid perhaps?

- **Authors:** *Yes, we indicated this in the citation: "The present-day annual mean sea surface salinity in the study area is 33.8 ±0.2 psu (World Ocean Atlas 2018, 0.25deg: Zweng et al., 2018)."*

RC2: Line 97: I think you are missing a word in this sentence. This should maybe read: "… of <2 ka, and a total of 27 samples were…" Also, you might consider noting how ODP 1234 was sampled (the same way?).

- **Authors:** *We added ",and". We did not sample ODP 1234, the sampling and analysis is described by de Bar et al., (2018) and Weiss et al., (2019b).*

RC2: Line 101: I think you meant to use a comma not a period to indicate 41,000 years.

- **Authors:** *Yes, thank you. We changed it.*

RC2: Line 103: Perhaps the word 'with' should be replaced with 'at' here.

- **Authors:** *Yes, thank you. We changed it.*

RC2: Line 113: Please define the average differences here – is this a standard error or standard deviation etc.?

- **Authors:** *We corrected it to standard deviation of replicate analysis of the same sample.*

RC2: Line 114: The standard deviation of what is within 0.1 per mil?

- **Authors:** *Both isotope calibration standards, NBS 19 and NFHS-1, are within 0.1‰ standard deviation of accepted values. We clarified it in the revised version.*

RC2: Line 119: Indicate the model number or instrument name for the ASE.

- **Authors:** *OK. We added Dionex 350 ASE.*

RC2: Line 143: RSME should be RMSE

- **Authors:** *Yes, thank you. We changed this.*

RC2: Line 146: Are you referring to the measured salinity in the modern surface open-ocean?

- **Authors:** *For the modern open-ocean waterline we included data from all depths. However, for the isotope to salinity calibration we report now in the manuscript: top 300 m water depth as we apply it to the surface water proxy $\delta H_{C37}$. We checked the dataset and statistics again and observed that the isotope-salinity calibration is not changing when using the whole dataset versus the top 300 m. We added the sampling depth information in the supplementary information.*

RC2: Line 149: Again, are you referring to surface ocean data?

- **Authors:** *The data points from the modern open-ocean dataset from include all sampling depths (between 0-4000 m).*

RC2: Line 154: Is there a good citation to include here for detailing that the oxygen isotopic composition of forams depends mainly on temperature and the isotopic composition of seawater?

- **Authors:** *In the lines following line 154 we do discuss publications regarding this matter, e.g. McCrea et al. (1950) being the first laboratory study showing the relationship of oxygen isotope signal of the foraminifera with that of growth water and temperature.*

RC2: Line 155: "...calcite sources..." is likely not the best/most correct term. Perhaps, calcite species or polymorphs?

- **Authors:** *We corrected it to polymorphs.*

RC2: Line 159: "... temperature proxies and **then** obtain salinity..."

- **Authors:** *We corrected it.*

RC2: Line 161: "... (2011)**, where** the $\delta^{18}$O- paleotemperature..."

- **Authors:** *We followed the reviewers suggestion.*

RC2: Line 162: in-situ should be italicized.

- **Authors:** *We changed it.*

RC2: Line 166: "...at the time of Eq. (4)..." is awkward wording, please considering clarifying.

- **Authors:** *We changed this.*

RC2: Line 167: Please define this equation, where is it from? As it stands it is not well introduced/prefaced.

- **Authors***: Thank you. We corrected this Line, it is referring to Eq. (5) and added another reference to Eq. 5.*

RC2: Line 169 (Equation 6): perhaps the authors might define earlier how temperature is solved for in this equation typically? Which proxy(ies) is/are used to resolve temperature.

- **Authors:** *We explained in the Introduction how the temperature is usually resolved, see line 39-43. We made an assumption for the temperature change, since we were not able to determine bottom water temperatures, as described in the Results and Discussion at lines 195-202.*

RC2: Line 190: "The oxygen **isotopic signatures** of benthic Uvigernia in ODP core 1235 range between 4.15 ..."

- **Authors:** *Thank you, we changed it.*

RC2: Line 192: $\delta^{18}$O$_{foram}$

- **Authors:** *We checked the abbreviations for consistency.*

RC2: Line 192: Perhaps change the word "till" to "to".

- **Authors:** *We changed it.*

RC2: Line 193: This sentence structure needs work. "...showing an LGM to 1 ka shift of..." is difficult for a reader to follow. Again, indicate the direction of the shift.

- **Authors:** *We changed it to: "This record is consistent with ODP core 1234, showing a negative isotope shift from the LGM to 1 ka of ca. 1.7‰ in the $\delta^{18}O$ values of Uvigerina (de Bar et al., 2018)."*

RC2: Line 195: This sentence structure could be improved.

- **Authors:** *We improved it.*

RC2: Line 201: What is the temperature change reported in ODP 1234 – if it is consistent, please report the value(s).

- **Authors:** *We changed these lines and report the temperature change clearer. It is consistent.*

RC2: Line 204: It is a little bit difficult to follow when the authors are referring to surface and benthic water isotope values – is there a clearer way to indicate benthic vs surface throughout?

- **Authors:** *We tried to make this clearer in the revised version by adding the abbreviations BSW, bottom seawater and SSW, surface seawater to the delta notations. Chapter 3.1 only describes bottom seawater isotopes as stated by the title and chapter 3.2 only surface seawater isotopes. We chose this structure to make the discussion clearer and help the reader. Chapter 3.3 is comparing the seawater isotopes, oxygen and hydrogen isotope proxies, and surface versus bottom water signals – here we tried to make clear when we speak of bottom and when of surface water. Chapter 3.4 discusses only the global surface water signal.*

RC2: Line 206: Please indicate the directionality of the isotopic shift.

- **Authors:** *Added the direction of the isotope shift.*

RC2: Line 218: '...globally surprisingly similar in magnitude," is not very clear wording.

- **Authors:** *We changed the wording.*

RC2: Figure 2: General introductory statement first. "**Alkenone** hydrogen isotopic ratios measured in ODP 1235....". What does 5pt average refer to? Where are the black circles on the figure indicating $\delta^{18}O$ of the benthic foram from ODP 1234? What are the seawater temperature data reconstructed from? TEX UK37?

- **Authors:** *We noticed that it needs to be: "The green line shows the 5pt average $\delta^{18}O$ values of benthic Foraminifera from ODP 1234 (from de Bar et al., 2019)". We edited the figure description accordingly.*

RC2: Line 246: Where was de Bar et al.'s work? Indicate this to the reader.

- **Authors:** We indicated in the figure caption first line that both core records are next to each other and from the Chilean Margin.

RC2: Line 262: I am not sure if it is commonplace to capitalize 'Recent'?

- **Authors:** *Yes, it is a noun in Geology; singular proper noun: Recent*

RC2: Line 269: salinity – $^2$H should be salinity- $\delta^2$H$_{sw}$ ?

- **Authors:** *We changed it to $\delta^2$H to avoid confusion.*

RC2: Line 271: "… of a dominating **evaporative regime** on the slope of the isotope…" and on which isotope are the authors referring? Both?

- **Authors:** *We are referring to both seawater isotopes. We followed the suggestion and added water-isotope.*

RC2: Line 273: "Surface waters may be more sensitive **to** changes in this…"

- **Authors:** *We changed this.*

RC2: Line 274: salinity – $^2$H should be salinity- $\delta^2$H$_{sw}$ ?

- **Authors:** *We changed it to $\delta^2$H to avoid confusion.*

RC2: Line 274: "… for surface waters, due to, **for example**…"

- **Authors:** *We changed this.*

RC2: Line 275-276: This sentence is very confusing to follow, please consider elaborating on the message here.

- **Authors:** *We have rephrased it in the manuscript.*

RC2: Table 1: Please consider adding a general statement to describe what this table is detailing. I would also consider being explicit about why the Kasper data are not included in your discussion.

- **Authors:** *We revised Table 1 substantially to make more clear what data is shown. We changed the table caption and state in the discussion that we compare the Chilean records to four global records which show a negative glacial-interglacial shift.*

RC2: Line 297-298: Please consider adjusting the "Globally distributed…" sentence as it is difficult to follow. Subscript C37 ($C_{37}$).

- **Authors:** *We adjusted the sentence and subscript $C_{37}$ throughout the revised manuscript.*